# CTC-DRO: Robust Optimization for Reducing Language Disparities in Speech Recognition

**Martijn Bartelds**[1][*]  **Ananjan Nandi**[1][*]  **Moussa Koulako Bala Doumbouya**[1]
**Dan Jurafsky**[1]  **Tatsunori Hashimoto**[1]  **Karen Livescu**[2]
[1]Department of Computer Science, Stanford University, Stanford, USA
[2]Toyota Technological Institute at Chicago, Chicago, USA

## Abstract

Modern deep learning models often achieve high overall performance, but consistently fail on specific subgroups. Group distributionally robust optimization (`group DRO`) addresses this problem by minimizing the worst-group loss, but it fails when group losses misrepresent performance differences between groups. This is common in domains like speech, where the widely used connectionist temporal classification (CTC) loss not only scales with input length but also varies with linguistic and acoustic properties, leading to spurious differences between group losses. We present `CTC-DRO`, which addresses the shortcomings of the `group DRO` objective by smoothing the group weight update to prevent overemphasis on consistently high-loss groups, while using input length-matched batching to mitigate CTC's scaling issues. We evaluate `CTC-DRO` on the task of multilingual automatic speech recognition (ASR) across five language sets from the diverse `ML-SUPERB 2.0` benchmark. `CTC-DRO` consistently outperforms `group DRO` and CTC-based baseline models, reducing the worst-language error by up to $47.1\%$ and the average error by up to $32.9\%$. `CTC-DRO` can be applied to ASR with minimal computational costs, and, while motivated by multilingual ASR, offers the potential for reducing group disparities in other domains with similar challenges.

## 1 Introduction

State-of-the-art deep learning models are often highly accurate on training data populations, while consistently underperforming on specific subpopulations or groups (Hashimoto et al., 2018; Duchi et al., 2023). One practical setting where this issue has very large effects is multilingual automatic speech recognition (ASR), where performance varies substantially between languages (Radford et al., 2023; Pratap et al., 2024; Shi et al., 2024). Such models, which jointly perform language identification (LID) and ASR in many languages, could help improve accessibility and increase digital participation for speakers worldwide (Besacier et al., 2014).

Distributionally robust optimization (`DRO`), particularly `group DRO` (Sagawa et al., 2020), has the potential to mitigate language disparities in multilingual ASR. `Group DRO` improves group robustness by up-weighting high-loss groups during training, and has been shown to outperform other approaches where the goal is to achieve high performance, even on the worst-performing group (Koh et al., 2021). However, it requires comparable training losses between groups to perform well (Oren et al., 2019; Sagawa et al., 2020), and this condition is often not met in ASR model training, because of differences in input length and speaker and acoustic characteristics across language-specific datasets.

In this paper, we focus on a training approach that has been successful on multilingual ASR benchmarks: pre-trained self-supervised models fine-tuned with the connectionist temporal classification (CTC; Graves et al., 2006) objective (Rouditchenko et al., 2023; Chen et al., 2024; Pratap et al., 2024). CTC-based models built on encoders such as `XLS-R` (Babu et al., 2022) and `MMS` (Pratap et al., 2024) are widely adopted and offer advantages over autoregressive models like `Whisper` (Radford et al., 2023), including faster inference and reduced hallucinations (Koenecke et al., 2024; Peng et al., 2024), which are crucial for many downstream applications. However, differences in CTC-based

---

[*]Equal contribution. Correspondence to `bartelds@stanford.edu`.

training losses due to length, speaker, and acoustics may lead to varying magnitudes and irreducible components of losses across different groups. As a result, the `group DRO` weights do not have the desired effect.

To address these issues, we present `CTC-DRO`, which optimizes a generalization of the `group DRO` objective, specifically by smoothing the up-weighting of high-loss groups. This new objective prevents overemphasis on groups with consistently and disproportionately high training losses. Also, we use length-matched group losses to mitigate the scaling properties of CTC. We evaluate `CTC-DRO` using language sets randomly selected from the `ML-SUPERB 2.0` (Shi et al., 2024) benchmark collection, which includes multilingual speech data from 15 diverse corpora across multiple domains, speaking styles and recording conditions. In this setting, `CTC-DRO` models outperform both `group DRO` and CTC-based baseline models across five language sets, regardless of whether balanced or unbalanced amounts of training data per language are used during training. Specifically, `CTC-DRO` models reduce the error rate of the worst-performing language in all of the five sets, with improvements of up to 47.1%, while also improving the average performance across all languages by up to 32.9%. While motivated by multilingual ASR, `CTC-DRO` offers the potential for reducing group disparities in other domains with incomparable training losses between groups, such as medical applications (Ganz et al., 2021; Petersen et al., 2023). Our code and newly trained models are publicly available at https://github.com/Bartelds/ctc-dro.

## 2 BACKGROUND

### 2.1 GROUP DRO

Given a family of models $\Theta$, loss function $\ell$ and training data $(x, y)$ drawn from empirical distribution $\hat{P}$, the standard training procedure for label prediction involves minimizing the expected loss over the training data:

$$\min_{\theta \in \Theta} \mathbb{E}_{(x,y) \sim \hat{P}} \left[ \ell(\theta; (x, y)) \right]. \tag{1}$$

In contrast, `group DRO` aims to minimize the worst-case expected loss over a set of pre-defined groups or sub-distributions $\{\hat{P}_g : g \in G\}$ in the training data:

$$\min_{\theta \in \Theta} \left\{ \max_{g \in G} \mathbb{E}_{(x,y) \sim \hat{P}_g} \left[ \ell(\theta; (x, y)) \right] \right\}. \tag{2}$$

Following Sagawa et al. (2020), this objective can be rewritten as:

$$\min_{\theta \in \Theta} \left\{ \sup_{q \in \Delta_{|G|}} \sum_{g \in G} q_g \mathbb{E}_{(x,y) \sim \hat{P}_g} \left[ \ell(\theta; (x, y)) \right] \right\}, \tag{3}$$

where $\Delta_{|G|}$ is the $|G|$-dimensional probability simplex, and $q_g$ is a weight for group $g \in G$. Sagawa et al. (2020) propose an online algorithm to optimize this objective, treating the problem as a minimax game and interleaving gradient ascent updates on $q = \{q_g : g \in G\}$ with gradient descent updates on $\theta$ for training data mini-batches (see Algorithm 2 in Appendix C).

### 2.2 CTC

The CTC objective (Graves et al., 2006) defines a method to learn a mapping between an input sequence $X = (x_1, x_2, \ldots, x_D)$ and an output sequence $Y = (y_1, y_2, \ldots, y_U)$ without requiring a known alignment between them, but assuming $U \leq D$ and a monotonic alignment. CTC uses a blank output token $\epsilon$ to handle $x_d \in X$ that do not map to any output symbol. Consider $\mathcal{Z}$, which is the set of all sequences of length $D$ that are composed of tokens from $Y$, and $\epsilon$. Each sequence $Z \in \mathcal{Z}$ is a potential alignment between $X$ and $Y$. CTC defines a collapsing function that merges consecutive, identical symbols and removes $\epsilon$ in an alignment $Z$. The set of alignments $Z \in \mathcal{Z}$ that collapse to $Y$ using this function forms the set of valid alignments $\mathcal{A}(X, Y)$. For example, a possible alignment $Z \in \mathcal{A}(X, Y)$ for $D = 2U + 2$ could be: $[\epsilon, y_1, \epsilon, y_2, y_2, \epsilon, \ldots, \epsilon, y_U, \epsilon]$. The conditional probability $P_{CTC}(Z|X)$ for any alignment $Z$ is computed as:

$$P_{CTC}(Z|X) = \prod_{d=1}^{D} p(z_d|X), \tag{4}$$

where $Z = (z_1, z_2, \ldots, z_D)$ and $p(z_d|X)$ is the model's predicted probability for symbol $z_d \in Z$ at time $d$. The predicted probability of the output sequence $Y$, $P_{CTC}(Y|X)$, is then computed by marginalizing over valid alignments $Z \in \mathcal{A}(X, Y)$:

$$P_{CTC}(Y|X) = \sum_{Z \in \mathcal{A}(X,Y)} P_{CTC}(Z|X). \tag{5}$$

The CTC loss function for $(X, Y)$ is then defined as:

$$\mathcal{L}_{CTC} = -\log P_{CTC}(Y \mid X). \tag{6}$$

## 2.3 Limitations of Group DRO Applied to CTC

The CTC loss, as defined in Equation 6, scales with the length of the input sequence $D$ and the length of the output sequence $U$. This scaling behavior occurs because $P_{CTC}(Y|X)$ is a marginalization over all valid alignments $Z \in \mathcal{A}(X, Y)$. Each alignment is a sequence of length $D$, which collapses to an output sequence of length $U$. As $D$ increases relative to $U$, the number of valid alignments increases as well (Graves et al., 2006). As each alignment's probability is the product of $D$ per-element probabilities, its value typically decreases as $D$ increases. Therefore, their sum $P_{CTC}(Y|X)$ remains relatively low, as the per-alignment probabilities typically decrease faster than the number of valid alignments increases. In practice, this often results in a higher CTC loss for longer sequences.

Therefore, differences in the distribution of $D$ or $U$ between groups can result in CTC losses that are not directly comparable. For example, a long audio sample (large $D$) may have fewer errors overall, but a higher loss than a short audio sample (small $D$) if their transcription lengths $U$ are similar. In Figure 1, we illustrate the need to address this challenge, showing that there are large differences in the distribution of audio sample lengths $D$ across various groups (in this case, languages) included in our experimental setup, which we further detail in Section 4. In this example, Spanish has a high proportion of long utterances, resulting in higher CTC losses. We find that the `group DRO` algorithm assigns a larger weight to this group, even though it is among the best groups in terms of downstream performance in our experiments, as shown in Section 5.

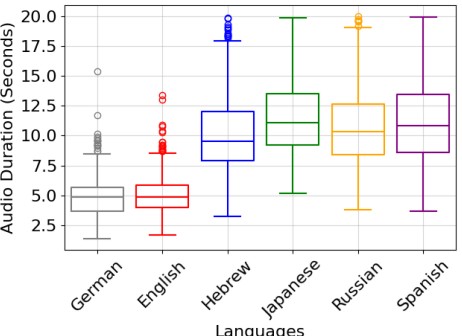

Figure 1: Distribution of audio sample lengths across groups (languages) in our experimental setup.

Importantly, simply scaling the CTC loss by $D$ or $U$ is insufficient to address the problem of incomparable CTC losses across languages (see Appendix G). In addition, the CTC loss also varies due to differences in linguistic and acoustic properties across the pre-defined groups. This may cause variance in the irreducible component of the training loss (Malinin & Gales, 2018).

In line with observations made in past work (Oren et al., 2019; Słowik & Bottou, 2022), we show that this inherent incomparability of losses across groups poses a critical challenge for `group DRO`. From Algorithm 2, we compute the gradient ascent update to $q_g$, given group losses $\mathcal{L}_g$, as:

$$q_g \leftarrow \frac{q_g \cdot \exp(\eta_q \mathcal{L}_g)}{\sum_g (q_g \cdot \exp(\eta_q \mathcal{L}_g))}. \tag{7}$$

This is equivalent to the Hedge algorithm (Slivkins, 2019) update for the following maximization objective:

$$\max_{q \in \Delta_{|G|}} \sum_{g \in G} q_g \mathcal{L}_g. \tag{8}$$

Now consider a situation where one of the groups $g'$ consistently has the highest training losses among all groups during training, presumably due to long audio samples or lengthy transcriptions, as well as the highest irreducible loss. This will result in its weight $q_{g'}$ consistently receiving the largest increases $\delta q_g$ during training, as:

$$\delta q_g \propto q_g \exp(\eta_q \mathcal{L}_g). \tag{9}$$

As a result, $q_{g'}$ will grow disproportionately large over the course of training, eventually drawing all the weight away from the other groups. This can result in other groups being under-weighted, which will cause a substantial decrease in their downstream performance (see Section 5).

This observation highlights the problems caused by the fundamental mismatch between the computed loss and the ideal loss for use in `group DRO`. The ideal loss would measure only the excess loss beyond each group's irreducible component and be length-normalized. However, in our setting, the irreducible component of the training loss is difficult to estimate, and, as we show in Appendix G, simple per-utterance scaling does not provide a solution. Existing solutions, such as calibrating group losses or approximating disparities between groups with simpler models (Oren et al., 2019; Słowik & Bottou, 2022), would either require a substantial increase in computational cost or a proxy for group difficulty, for which there is no reliable model for speech to our knowledge. Therefore, CTC remains inherently incompatible with `group DRO`.

## 3 CTC-DRO

To address the identified challenges, we propose a new training algorithm: `CTC-DRO` (Algorithm 1). This algorithm computes length-matched losses across groups to mitigate the scaling properties of CTC, and uses a generalization of the `group DRO` objective that introduces a new smoothed maximization objective for the group weights to prevent overemphasis on groups with consistently high training losses. Like `group DRO`, `CTC-DRO` has minimal computational costs, only keeping track of a single scalar weight for every group in the training data.

### 3.1 LENGTH-MATCHED GROUP LOSSES

To address incomparable CTC losses across groups due to different distributions of audio lengths, we ensure that the CTC loss for each group is computed using roughly equal total audio durations. Specifically, we create a new batch sampler that selects batches of audio samples and corresponding transcripts $(x_i, y_i)$, all from a single, randomly-selected group $g$, while ensuring that their total audio duration is as close to a fixed value (set as a hyperparameter) as possible.[1] Batches with a larger number of shorter audio samples tend to have a lower CTC loss per audio sample than batches with fewer, longer, audio samples. Therefore, we sum the utterance-level CTC losses in a batch (see line 10 in Algorithm 1) and update the group weights using this sum instead of the mean loss used in the `group DRO` algorithm. During training, these summed losses are tracked for each group, and a group weight update is performed only after at least one batch has been processed for every group. If a group is sampled multiple times before the update, the corresponding summed losses are averaged. This approach effectively increases the batch size for computing the group weight update.

Also, we multiply the losses by the total number of groups (line 21 in Algorithm 1) before performing gradient descent on the model parameters. This ensures that the training losses

---

**Algorithm 1** Optimization algorithm for `CTC-DRO`. $\theta$ represents the model parameters.

1: **Input:** Step sizes $\eta_q, \eta_\theta$; smoothing parameter $\alpha$; loss function $\ell$; duration of each batch $d$; groups $G = \{g\}$; training data $(x, y, g) \sim D$; number of training steps $T$
2: Initialize $\theta^{(0)}, \{q_g\}$
3: Initialize gr_losses[g] $= \emptyset \ \forall g$
4: **for** $t = 1$ to $T$ **do**
5:     Sample $g \sim G$
6:     Sample $\mathcal{B} = \{(x_i, y_i, g)\}_{i=1}^{B_t} \sim D$   // selected such that $\Sigma_{i=1}^{B_t} \text{duration}(x_i) \approx d$
7:     **for** $i = 1$ to $B_t$ **do**
8:         $\ell_i = \ell(\theta^{(t-1)}; (x_i, y_i))$
9:     **end for**
10:    gr_losses[g] $\leftarrow$ gr_losses[g] $\cup \left\{ \sum_{i=1}^{B_t} \ell_i \right\}$
11:    **if** gr_losses[g] $\neq \emptyset \ \forall g$ **then**
12:       **for each** group $g$ **do**
13:          $\bar{\ell}_g = \dfrac{\sum_{\mathcal{L} \in \text{gr\_losses}[g]} \mathcal{L}}{|\text{gr\_losses}[g]|}$
14:          $q'_g \leftarrow q_g \times \exp\left( \dfrac{\eta_q \bar{\ell}_g}{q_g + \alpha} \right)$
15:          gr_losses[g] $\leftarrow \emptyset$
16:       **end for**
17:       **for each** group $g$ **do**
18:          $q_g \leftarrow \dfrac{q'_g}{\sum_{g'} q'_{g'}}$     // gradient ascent on $q$
19:       **end for**
20:    **end if**
21:    $\tilde{\mathcal{L}} = q_g |G| \sum_{i=1}^{B_t} \ell_i$ // all data from same group $g$
22:    $\theta^{(t)} \leftarrow \theta^{(t-1)} - \eta_\theta \nabla_\theta \tilde{\mathcal{L}}$ // gradient descent on $\theta$
23: **end for**

---

[1] Group utterances are iteratively added to a batch until the total duration meets or slightly exceeds the set target duration.

with `CTC-DRO` are comparable to a model trained without `CTC-DRO`, removing the need to tune shared hyperparameters, such as the learning rate, separately for both training algorithms.

## 3.2 SMOOTHED MAXIMIZATION OBJECTIVE

We propose a new method for updating the group weights, which addresses `group DRO`'s tendency to assign a disproportionately large weight to groups with consistently high training losses (see Section 2.3). This approach also helps mitigate the scaling properties of CTC related to transcription length, which cannot be adequately resolved by normalizing for transcript length (see Appendix G).

Our proposed update rule introduces a smoothing hyperparameter $\alpha$ (see Algorithm 1):

$$q_g \leftarrow \frac{q_g \cdot \exp(\eta_q \frac{\mathcal{L}_g}{q_g + \alpha})}{\sum_{g \in G}(q_g \cdot \exp(\eta_q \frac{\mathcal{L}_g}{q_g + \alpha}))}. \tag{10}$$

As $\alpha \to 0$, the update becomes increasingly more sensitive to the current group weight relative to the group loss. This causes groups with higher weights to receive smaller updates, resulting in a more uniform distribution of the group weights. In contrast, as $\alpha$ increases, updates depend more on the group loss compared to the group weight, increasing the group weights more strongly for groups with higher losses. In fact, when $\alpha \to \infty$, the update rule reduces to:

$$q_g \leftarrow \frac{q_g \cdot \exp(\eta_q \frac{\mathcal{L}_g}{\alpha})}{\sum_{g \in G}(q_g \cdot \exp(\eta_q \frac{\mathcal{L}_g}{\alpha}))}, \tag{11}$$

recovering the form of the `group DRO` update.

This update rule has several desirable properties. First, the updates to $q_g$ are smoother, because they are inversely proportional to the current $q_g$, while still being proportional to the loss $\mathcal{L}_g$. This discourages any single group from having a disproportionately large weight $q_g$ relative to its group loss, leading to a more balanced distribution of the group weights. Second, the update rule adjusts for differences in group weights when the CTC losses are similar. Specifically, if two groups with different $q_g$ have similar CTC losses, the group with the lower $q_g$ receives a larger update. This helps prevent under-training of lower-weighted groups by reducing the gap between the group weights over time.

Along with these desirable properties, we demonstrate that our new objective does not change the fundamental behavior of the `group DRO` objective, assigning higher weights to groups with higher losses. Following the Hedge algorithm (Slivkins, 2019), Equation 10 optimizes the following generalization of the `group DRO` maximization objective (Equation 8):

$$\max_{q \in \Delta_{|G|}} \sum_{g \in G} \log(q_g + \alpha)\mathcal{L}_g. \tag{12}$$

Expanding the conditions for the probability simplex $\Delta_{|G|}$ ($\sum_g q_g = 1$, $q_g \geq 0 \ \forall g$) and taking the Lagrangian of Equation 12, we obtain:

$$\mathcal{J} = \sum_{g \in G} \log(q_g + \alpha)\mathcal{L}_g + \lambda(1 - \sum_{g \in G} q_g) - \sum_{g \in G} \lambda_g q_g, \tag{13}$$

where $\lambda$ and $\lambda_g$ are Lagrange multipliers and $\lambda_g \geq 0$ for all $g$. To find the optimal $q_g$, we calculate the partial derivative of $\mathcal{J}$ with respect to $q_g$ and set it to 0:

$$\frac{\partial \mathcal{J}}{\partial q_g} = \frac{\mathcal{L}_g}{q_g + \alpha} - \lambda - \lambda_g = 0 \quad \Longrightarrow \quad q_g = \frac{\mathcal{L}_g}{\lambda + \lambda_g} - \alpha. \tag{14}$$

Assuming $q_g > 0$ for all $g$, complementary slackness ($\lambda_g q_g = 0$ with $\lambda_g \geq 0$ for all $g$) implies $\lambda_g = 0$ for all $g$ and:

$$q_g = \frac{\mathcal{L}_g}{\lambda} - \alpha. \tag{15}$$

Since $\sum_g q_g = 1$:

$$1 = \sum_g (\frac{\mathcal{L}_g}{\lambda} - \alpha) \quad \Longrightarrow \quad \lambda = \frac{\sum_g \mathcal{L}_g}{1 + |G|\alpha} \tag{16}$$

Substituting in Equation 15:

$$q_g = \frac{\mathcal{L}_g(1 + |G|\alpha)}{\sum_g \mathcal{L}_g} - \alpha \quad \implies \quad q_g + \alpha \propto \frac{\mathcal{L}_g}{\sum_{g'} \mathcal{L}_{g'}} \tag{17}$$

Thus, the optimal weight for a group ($q_g$) increases with its loss ($\mathcal{L}_g$), since $q_g + \alpha$ is proportional to $\mathcal{L}_g$ and both $\alpha$ and $\sum_{g'} \mathcal{L}_{g'}$ are constant with respect to $g$.

## 4 EXPERIMENTS

We fine-tune the existing, self-supervised multilingual `XLS-R` and `MMS` models on the task of multilingual ASR (formulated as a joint task of ASR and LID) using data from the `ML-SUPERB 2.0` benchmark (more on the dataset in Section 4.1). These models are licensed under Apache 2.0 and CC-BY-NC-4.0, respectively. Following the setup of `ML-SUPERB 2.0`, we add two Transformer layers and a softmax layer on top of the pre-trained models to predict a language token followed by character sequences using CTC. We do not use a separate LID head or loss, and update all model parameters. The models we choose have shown the best performance on `ML-SUPERB 2.0` (Shi et al., 2024), outperforming other models like `Whisper` (Radford et al., 2023). The two pre-trained models share the same architecture and training objective (Baevski et al., 2020), but their training data differs: `XLS-R` is pre-trained on roughly 436K hours of speech in 128 languages, while `MMS` is pre-trained on 491K hours of speech in 1406 languages.

We train the models using three approaches. First, our baseline models use the joint ASR and LID training setup adopted in `ML-SUPERB 2.0` (as described above), with the addition of our new batch sampler that computes length-matched group losses. Second, we fine-tune models using our proposed `CTC-DRO` algorithm. Third, we train models using the `group DRO` algorithm (replicating its original batch sampler) for comparison. When training both `CTC-DRO` and `group DRO` models, the groups correspond to the languages in our training datasets (see Section 4.1).

We mostly follow the hyperparameters used by Babu et al. (2022), Pratap et al. (2024), and in `ML-SUPERB 2.0`, but train for 40 epochs, retaining the model checkpoint with the lowest loss on the development data, accumulate gradients across 16 batches, set the batch duration hyperparameter (Algorithm 1) so that batches fit within our NVIDIA A6000 GPU memory, leading to batches of roughly 50 seconds of audio (more details in Appendix F), and tune the learning rate of the baseline models on our development data. We also use this learning rate to train models with `CTC-DRO` and `group DRO`. Lastly, for the `CTC-DRO` and `group DRO` models, we tune the DRO-specific hyperparameters on the development set as well, specifically $\eta_q \in \{10^{-3}, 10^{-4}\}$ and $\alpha \in \{0.1, 0.5, 1\}$.

### 4.1 DATASET

We use the `ML-SUPERB 2.0` dataset for our experiments. This dataset belongs to an established benchmark where a number of multilingual ASR models have already been compared. It has broad coverage of 141 languages sourced from 15 corpora, and contains substantial variation in domains and recording environments as well as more natural speech compared to smaller, translation focused corpora, such as `FLEURS` (Conneau et al., 2023). For each language-corpus pair, there is between one and nine hours of training data available, as well as 10 minutes each for development and test data. While we focus on studying relatively small training data sizes, prior work has shown that ASR performance differences between languages persist even when the amount of training data increases substantially (e.g., see Radford et al., 2023).

For our main experiments, we use a balanced data setup by randomly selecting five diverse sets of groups from `ML-SUPERB 2.0`, each consisting of six language-corpus pairs, matching the number of groups used in Sagawa et al. (2020). We thus have one hour of training data, and 10 minutes of development and test data available for each language-corpus pair in each set. The selection of language-corpus pairs is based on the character error rates (CERs) of the best-performing model configuration from `ML-SUPERB 2.0`. Specifically, for each set, we randomly select two language-corpus pairs from the bottom 10 percentile of CERs, two language-corpus pairs from the top 10 percentile of CERs, and two language-corpus pairs with CERs between the 10th and 90th percentiles.

For the first two language sets, we also investigate the effect of using additional training data in an unbalanced setup, as most languages in these sets have more than one hour of training data available. We show more dataset details in Appendix D.

## 4.2 EVALUATION

We compare the performance of `CTC-DRO` models to the baseline and `group DRO` models. They are evaluated using the standard CER metric on the test sets from the five language sets (metric details in Appendix E). We also report the LID accuracy for completeness. We report the CER of the worst-performing language (our primary metric), as well as the average CER across languages. For the `CTC-DRO` and `group DRO` models, we report the performance of the model checkpoint with the largest CER improvement on the worst-performing language relative to the baseline on the development set.

## 5 RESULTS

We present the results of our experiments using balanced and additional training data in Table 1 and Table 2, respectively (detailed results, including hyperparameter search results and a word error rate (WER) analysis, in Appendix F; wall-clock training times in Appendix I). In line with previous work (e.g., Pratap et al., 2024 and Shi et al., 2024), we find substantial performance differences between languages for our baseline models trained without `group DRO` or `CTC-DRO`, as shown by the large difference between the CER of the worst-performing language and the average CER across languages. This finding applies to each of the evaluated sets, regardless of whether the training data is balanced or unbalanced across languages.

Table 1: CER of the worst-performing language (`Max CER`, ISO code for the worst-performing language provided as `ISO`), as well as the average CER (`Avg CER`) and LID accuracy (`LID`) across languages (in %) for the baseline models (`Base`), `group DRO` models (`GDRO`), and `CTC-DRO` models (`Ours`) on the test sets from the five language sets (indexed by the "#" column). Best results are highlighted.

| Set # | Model | Type | $\eta_q$ | $\alpha$ | Max CER (ISO) ($\downarrow$) | Avg CER ($\downarrow$) | LID ($\uparrow$) | Set # | Model | Type | $\eta_q$ | $\alpha$ | Max CER (ISO) ($\downarrow$) | Avg CER ($\downarrow$) | LID ($\uparrow$) |
|---|---|---|---|---|---|---|---|---|---|---|---|---|---|---|---|
| 1 | MMS | Base | | | 60.8 (NAN) | 23.4 | **97.4** | 2 | MMS | Base | | | 49.4 (YUE) | 15.8 | **98.4** |
| | | GDRO | $10^{-4}$ | | 86.6 (NAN) | 30.5 | 78.7 | | | GDRO | $10^{-4}$ | | 55.5 (YUE) | 20.7 | 98.2 |
| | | Ours | $10^{-4}$ | 1.0 | **56.8** (NAN) | **22.9** | 95.8 | | | Ours | $10^{-3}$ | 0.5 | **44.4** (YUE) | **15.0** | 96.2 |
| | XLS-R | Base | | | 64.9 (CMN) | 25.2 | **92.6** | | XLS-R | Base | | | 68.8 (YUE) | 19.0 | **94.2** |
| | | GDRO | $10^{-4}$ | | 78.4 (NAN) | 30.0 | 87.8 | | | GDRO | $10^{-4}$ | | 58.8 (YUE) | 21.6 | 87.0 |
| | | Ours | $10^{-4}$ | 0.1 | **57.6** (NAN) | **22.5** | 89.5 | | | Ours | $10^{-4}$ | 0.5 | **45.0** (YUE) | **15.8** | 89.3 |
| 3 | MMS | Base | | | 34.2 (KOR) | 16.1 | 98.5 | 4 | MMS | Base | | | 24.0 (SND) | 14.4 | 87.9 |
| | | GDRO | $10^{-4}$ | | 34.0 (KOR) | 22.0 | **98.7** | | | GDRO | $10^{-4}$ | | 21.8 (URD) | 14.9 | **91.9** |
| | | Ours | $10^{-4}$ | 0.1 | **31.3** (KHM) | **15.3** | **98.7** | | | Ours | $10^{-3}$ | 0.5 | **18.4** (URD) | **12.9** | 87.3 |
| | XLS-R | Base | | | 33.2 (KHM) | **17.0** | 99.2 | | XLS-R | Base | | | 29.7 (URD) | 14.6 | 88.4 |
| | | GDRO | $10^{-4}$ | | 38.0 (KHM) | 25.1 | 97.2 | | | GDRO | $10^{-3}$ | | 25.6 (SLV) | 18.6 | 83.5 |
| | | Ours | $10^{-4}$ | 0.1 | **32.2** (KHM) | 17.7 | 97.9 | | | Ours | $10^{-3}$ | 0.1 | **24.2** (URD) | **13.7** | **88.9** |
| 5 | MMS | Base | | | 90.0 (JPN) | 26.0 | **96.3** | | | | | | | | |
| | | GDRO | $10^{-4}$ | | 62.2 (JPN) | 29.2 | 67.0 | | | | | | | | |
| | | Ours | $10^{-3}$ | 1.0 | **57.5** (JPN) | **24.3** | 90.5 | | | | | | | | |
| | XLS-R | Base | | | 114.8 (JPN) | 29.9 | 89.0 | | | | | | | | |
| | | GDRO | $10^{-4}$ | | 92.9 (JPN) | 36.8 | 57.7 | | | | | | | | |
| | | Ours | $10^{-4}$ | 0.1 | **71.5** (JPN) | **23.8** | **91.0** | | | | | | | | |

For each language set, `CTC-DRO` models achieve a lower CER for the worst-performing language compared to the baseline and `group DRO` models. The largest improvement is obtained on set 2 using `XLS-R` using all available data, showing a relative CER reduction of 47.1% compared to the baseline model. Note that `CTC-DRO` also results in the best average CER in 13 out of 14 settings (seven sets with two models each) compared to both the baseline and `group DRO` models, leading to relative CER reductions up to 32.9%. The exception is `XLS-R` in balanced set 3, where the average CER is slightly worse with `CTC-DRO` (17.7%) than the baseline (17.0%). In terms of LID accuracy, `CTC-DRO` models improve over the baseline models in seven out of 14 settings. In most of the remaining settings, the LID accuracy of `CTC-DRO` models exceeds 95%, leaving little room for

further improvement. To assess sensitivity to random initialization, we report the performance of baseline and `CTC-DRO` models on sets 1 and 3, which have the smallest single-seed worst-language improvements, with four different random seeds in Appendix F. The results show that the largest gains in worst-language CER are stable across seeds.

In contrast, `group DRO` worsens the CER of the worst-performing language in seven out of 14 settings compared to the baseline model, with the highest relative CER increase of 57.5% on set 2 using `MMS` trained on all available training data. Also, `group DRO` increases the average CER compared to the baseline in all settings. This finding shows the ineffectiveness of the original `group DRO` formulation in this challenging setting, and the substantial added robustness of the modifications in `CTC-DRO`.

In four settings, the worst-performing language changes between the baseline and `CTC-DRO` models. For example, in set 3 with `MMS` trained on balanced data, it shifts from Korean to Khmer. As shown in Table 9, the `CTC-DRO` model reduces the CER for Korean from 34.2 to 27.6, while the CER for Khmer remains unchanged at 31.3. Overall, `CTC-DRO` consistently improves the performance on the worst-performing language without significantly worsening best-language performance, while still achieving a lower CER on average (see Appendix F for detailed results and best-language analysis).

Table 2: CER of the worst-performing language (`Max CER`, ISO code for the worst-performing language provided as `ISO`), as well as the average CER (`Avg CER`) and LID accuracy (`LID`) across languages (in %) for the baseline models (`Base`), `group DRO` models (`GDRO`), and `CTC-DRO` models (`Ours`) on the test sets from the first two language sets using additional training data if available. Best results are highlighted.

| Set # | Model | Type | $\eta_q$ | $\alpha$ | Max CER (ISO) ($\downarrow$) | Avg CER ($\downarrow$) | LID ($\uparrow$) | Set # | Model | Type | $\eta_q$ | $\alpha$ | Max CER (ISO) ($\downarrow$) | Avg CER ($\downarrow$) | LID ($\uparrow$) |
|---|---|---|---|---|---|---|---|---|---|---|---|---|---|---|---|
| 1 | MMS | Base | | | 67.5 (NAN) | 25.6 | 98.1 | 2 | MMS | Base | | | 66.9 (YUE) | 19.5 | 99.0 |
| | | GDRO | $10^{-4}$ | | 96.3 (NAN) | 37.8 | 83.9 | | | GDRO | $10^{-3}$ | | 105.4 (YUE) | 38.8 | 81.0 |
| | | Ours | $10^{-4}$ | 0.5 | **62.8** (NAN) | **22.8** | **98.5** | | | Ours | $10^{-4}$ | 1.0 | **48.1** (YUE) | **16.4** | **99.1** |
| | XLS-R | Base | | | 92.1 (CMN) | 35.6 | 96.4 | | XLS-R | Base | | | 97.2 (YUE) | 28.0 | 98.2 |
| | | GDRO | $10^{-4}$ | | 90.8 (NAN) | 38.1 | 72.3 | | | GDRO | $10^{-4}$ | | 102.9 (YUE) | 44.0 | 80.8 |
| | | Ours | $10^{-4}$ | 1.0 | **67.5** (NAN) | **26.9** | **97.1** | | | Ours | $10^{-4}$ | 1.0 | **51.4** (YUE) | **18.8** | **98.6** |

# 6 ANALYSIS

Next, we present an ablation study to measure the contributions of the length-matched group losses and smoothed maximization objective introduced in `CTC-DRO` (Section 6.1). To this end, we train `CTC-DRO` models with each of these components removed one at a time on balanced training data from set 5, which showed the largest reduction in CER for the worst-performing language (Table 1). We also describe and compare how the group weights of `CTC-DRO` and `group DRO` models change throughout training (Section 6.2). For brevity, we focus on the `XLS-R` models trained on the same set, showing that `CTC-DRO` results in more stable training. Finally, we confirm the benefit of `CTC-DRO` when scaling to a larger number of groups (Section 6.3).

Table 3: CER of the worst-performing language (`Max CER`), as well as the average CER (`Avg CER`) and LID accuracy (`LID`) across languages (in %) on set 5 for a subtractive ablation of `CTC-DRO` (`Ours`), removing the length-matched group losses (`Dur`) and smoothed maximization objective (`Smooth`). Baseline (`Base`) results are shown for reference. Best results are highlighted.

| Model | Type | Max CER (ISO) ($\downarrow$) | Avg CER ($\downarrow$) | LID ($\uparrow$) |
|---|---|---|---|---|
| MMS | Base | 90.0 (JPN) | 26.0 | **96.3** |
| | Ours | **57.5** (JPN) | **24.3** | 90.5 |
| | - Dur | 84.6 (JPN) | 29.6 | 66.1 |
| | - Smooth | 102.1 (JPN) | 97.9 | 13.2 |
| XLS-R | Base | 114.8 (JPN) | 29.9 | 89.0 |
| | Ours | **71.5** (JPN) | **23.8** | **91.0** |
| | - Dur | 115.2 (JPN) | 50.6 | 54.4 |
| | - Smooth | 194.2 (JPN) | 61.4 | 43.2 |

## 6.1 ABLATION STUDY

We find that removing either component from `CTC-DRO` leads to a substantial decrease in performance (see Table 3; we present detailed results in Appendix F). Specifically, the CER of the

worst-performing language increases by up to 171.6% and the average CER by up to 302.9% compared to a model trained using the complete `CTC-DRO` algorithm. We also find that the smoothed maximization objective has the stronger effect on reducing both the CER of the worst-performing language and the average CER. Note that removing the smoothed maximization objective from `CTC-DRO` is not similar to training baseline models, as this configuration still uses the `group DRO` weight update mechanism (see Appendix C).

## 6.2 COMPARISON OF GROUP WEIGHTS

To analyze the behavior of `group DRO` and `CTC-DRO` models during training, we plot the group weights for all languages in set 5 throughout training in Figure 2 (see Appendix F for additional plots). The group weights of the `group DRO` model fluctuate substantially during training, reaching values close to 0 or 1 at various stages of training. For extended periods of training with `group DRO`, the group weights are heavily concentrated on a single language, causing the weights for all other languages to reach values close to 0.

In contrast, the group weights of the `CTC-DRO` model are distributed more evenly across all languages throughout training. The group weights for each language also fluctuate substantially less during training compared to `group DRO`. The only languages with group weights dropping below 0.1 at any point are German and English, both of which have low CERs on the development set. Importantly, the weight of Japanese (worst-performing) consistently remains among the top two highest group weights.

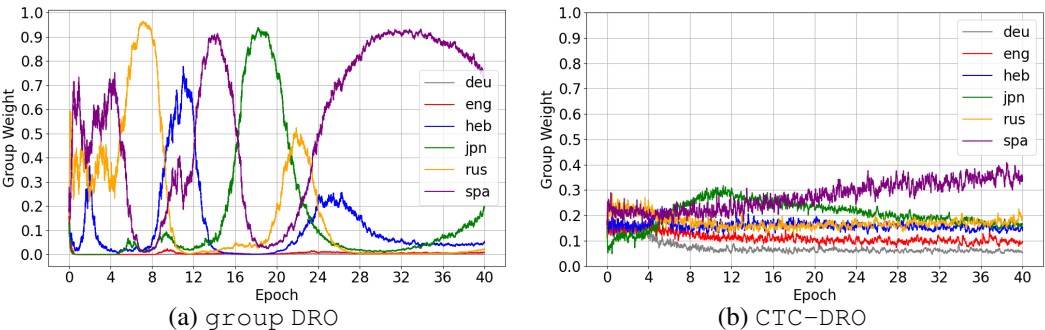

(a) `group DRO`        (b) `CTC-DRO`

Figure 2: Group weights for each language throughout training of an `XLS-R` model trained with `group DRO` or `CTC-DRO` on balanced data from set 5.

## 6.3 SCALABILITY TO MORE GROUPS

To analyze the impact of scaling the number of languages, we conduct additional experiments on 18 languages (our languages from set 1 plus 12 randomly sampled extra languages). We find that `CTC-DRO` maintains its effectiveness at improving worst-language performance, reducing the worst-language CER by 8.9% for `MMS` and 9.2% for `XLS-R` in the balanced data setting compared to baseline models. In the unbalanced data setting, `XLS-R` shows the strongest results with a relative CER reduction of 23.7% on the worst-performing language. The full results are shown in Appendix H.

## 7 RELATED WORK

**Robustness to distribution shifts** Prior work categorizes distribution shifts as domain generalization (Quiñonero-Candela et al., 2008; Hendrycks et al., 2021; Santurkar et al., 2021), where train and test data domains have no overlap, or subpopulation shifts (Dixon et al., 2018; Oren et al., 2019; Sagawa et al., 2020), where train and test data come from the same domains, but do not necessarily appear in the same proportions (Koh et al., 2021). Our experimental setup is an example of a subpopulation shift, as all test languages are included in the training data for the models.

Methods for robust generalization are commonly categorized into three groups. Domain invariance methods aim to learn feature representations that are consistent across domains (groups) by encouraging similar feature distributions across domains (Tzeng et al., 2014; Long et al., 2015; Ganin et al., 2016; Sun & Saenko, 2016). Other approaches use invariant prediction methods from the causal

inference literature (Meinshausen & Bühlmann, 2015; Peters et al., 2016; Arjovsky et al., 2019; Rothenhäusler et al., 2021). In contrast, `DRO` explicitly minimizes the worst-case loss over an uncertainty set, which is typically defined as a divergence ball around the training distribution (Namkoong & Duchi, 2016; Bertsimas et al., 2018; Esfahani & Kuhn, 2018; Duchi & Namkoong, 2019; Oren et al., 2019; Sagawa et al., 2020). Our work builds upon `group DRO` (Sagawa et al., 2020), since it has outperformed other approaches in settings with subpopulation shifts (Koh et al., 2021).

**Robust ASR**    Prior work on robustness in ASR primarily focuses on quantifying or addressing biases related to accent, age, dialect, gender, and race  (Tatman, 2017; Koenecke et al., 2020; Markl, 2022; Martin & Wright, 2022; Ngueajio & Washington, 2022; Feng et al., 2024; Harris et al., 2024). Methods to mitigate these biases include data balancing (Dheram et al., 2022) and fairness-promoting training methods (Sarı et al., 2021; Zhang et al., 2022; Veliche & Fung, 2023). These methods are not appropriate for reducing ASR language disparities, as they require large amounts of training data unavailable for most languages or have methodological constraints that prohibit direct application to a multilingual setting. Alternative approaches focused on multilingual settings use architectural and representation level improvements to include language information (Chen et al., 2023; Lu et al., 2024). These methods improve multilingual ASR performance by conditioning the model on language identity through auxiliary CTC objectives or conditional adapters. `CTC-DRO` differs in its objective, directly targeting worst-group performance through robust optimization rather than architectural modifications, but could in principle be combined with such approaches. Gao et al. (2022) explored `DRO` for training language-independent speech recognition models, and reported negative results.

**Comparison with other approaches**    We consider several alternative approaches but find them unsuitable for our multilingual ASR setting. For approaches that calibrate group losses or approximate disparities with simpler models (Oren et al., 2019; Słowik & Bottou, 2022), Section 2.3 explains that they would require substantially more computation or a proxy for group difficulty, for which there is no reliable model for speech. For other DRO variants that update on group-averaged losses (e.g., Lokhande et al., 2022), CTC losses remain not directly comparable across groups (see Section 2.3), and loss normalization does not solve this problem (as shown in Appendix G). Alternatively, group-aware reinforcement learning methods (e.g., Tjandra et al., 2018) could be used, but decoding during training and optimizing a sequence-level reward such as CER would be substantially more expensive than the scalar group-weight update used by `CTC-DRO`. To the best of our knowledge, our work is the first to propose a robust optimization method that successfully reduces cross-lingual performance disparities in ASR.

## 8    CONCLUSION

`CTC-DRO`, our robust optimization approach motivated by multilingual ASR, addresses `group DRO`'s inability to handle group losses that do not accurately reflect performance differences between groups. When applied to data from an established multilingual ASR and LID benchmark, `CTC-DRO` outperformed baseline CTC-based and `group DRO` models, reducing the worst-language CER across all sets and improving average CER and LID accuracy in almost all cases. Our analysis showed that this result can be attributed to the smoothed maximization objective and length-matched batching that balance and stabilize the group weights.

While performance disparities are reduced in our approach, they are not eliminated. The improvements may be sufficient to make ASR useful for more languages than before, but additional work is needed before ASR is truly practical for many more languages. A promising direction for future work is to automatically learn data groupings, which removes the need for pre-defined groups that may be unknown or incomplete, as well as applying `CTC-DRO` to pre-training. Extending `CTC-DRO` to code-switching scenarios is another promising direction (e.g., see Liu et al., 2024).

Also, we believe the principles underlying `CTC-DRO` have broader applicability. The smoothed maximization objective could in principle be applied to any setting with group-level losses, suggesting potential extensions to other architectures, loss functions, and groupings. For example, tasks that use variable-length sequences as input data and therefore face similar challenges, such as text classification and video transcription, could potentially benefit from our algorithm, enabling more inclusive processing of other data modalities as well.

ACKNOWLEDGMENTS

The authors thank Jiatong Shi for valuable discussions on the ESPnet toolkit and Jose Cruzado for helpful feedback on the algorithm description. We furthermore thank the anonymous reviewers for their insightful comments, which have allowed us to improve this paper.

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

## A   IMPACT STATEMENT

Our `CTC-DRO` approach reduces performance differences between languages in modern multilingual ASR models with minimal computational costs, ensuring it can be readily adopted. Our work therefore has the potential to positively impact speakers of many languages worldwide, including digitally underrepresented languages and varieties, by improving their access to information and services. However, several challenges remain. The performance of multilingual ASR needs to further improve before it can be deployed in real-world settings for many languages. In addition, improved ASR for underrepresented languages and varieties calls for careful, community-driven evaluation to ensure modern technology is aligned with local interests. In this process, it is important to evaluate `CTC-DRO`'s impact in varied real-world settings to ensure our algorithm benefits all language communities.

## B   REPRODUCIBILITY STATEMENT

To ensure the reproducibility of our work, we provide detailed descriptions of our algorithm and experimental setup. Specifically, the theoretical formulation of `CTC-DRO` is presented in Section 3. A comprehensive overview of our experimental framework, including datasets, model configurations, hyperparameter selection process, and evaluation setup is presented in Section 4. The experiments were performed on the publicly available `ML-SUPERB 2.0` benchmark, and we provide the exact information needed to reconstruct the language sets used in our experiments in Appendix D. Our code and newly trained models are publicly available at `https://github.com/Bartelds/ctc-dro`.

## C   GROUP DRO ALGORITHM

In Section 2.1, we described `group DRO`. Sagawa et al. (2020) propose an online algorithm to optimize the `group DRO` objective, which we show in Algorithm 2. They treat the optimization problem as a minimax game and interleave gradient ascent updates on $q = \{q_g : g \in G\}$ with gradient descent updates on $\theta$ for training data mini-batches.

---

**Algorithm 2** Online optimization algorithm for `group DRO`. $\theta$ represents the model parameters.

---

1: **Input:** Step sizes $\eta_q, \eta_\theta$; loss function $\ell$; batch size $B$; groups $G = \{g\}$; training data $(x, y, g) \sim D$; number of training steps $T$
2: Initialize $\theta^{(0)}$ and $\{q_g\}$
3: **for** $t = 1$ to $T$ **do**
4:     Sample $\mathcal{B} = \{ (x_i, y_i, g_i) \}_{i=1}^B \sim D$
5:     **for** $g \in G$ **do**
6:         $\mathcal{L}_g \leftarrow \emptyset$
7:         **for** $i = 1$ to $B$ **do**
8:             **if** $g_i == g$ **then**
9:                 $\mathcal{L}_g \leftarrow \mathcal{L}_g \cup \{\ell(\theta^{(t-1)}; (x_i, y_i))\}$
10:            **end if**
11:        **end for**
12:        $\bar{\mathcal{L}}_g = \dfrac{\sum_{\mathcal{L} \in \mathcal{L}_g} \mathcal{L}}{|\mathcal{L}_g|}$
13:        $q'_g \leftarrow q_g \exp(\eta_q \bar{\mathcal{L}}_g)$
14:    **end for**
15:    **for** $g \in G$ **do**
16:        $q_g \leftarrow \dfrac{q'_g}{\sum_{g'} q'_{g'}}$                                    // gradient ascent on $q$
17:    **end for**
18:    $\mathcal{L} \leftarrow \sum_{g \in G} q_g \bar{\mathcal{L}}_g$
19:    $\theta^{(t)} \leftarrow \theta^{(t-1)} - \eta_\theta \nabla_\theta \mathcal{L}$                            // gradient descent on $\theta$
20: **end for**

---

# D  DATASETS

In Table 4, we show the language-corpus pairs included in our main experiments. In Table 5, we show the number of samples, along with the average duration and transcript length for each language in each language set. Table 6 shows the first two language sets, listing all available corpora for each language in `ML-SUPERB 2.0`. All corpora in `ML-SUPERB 2.0` are licensed under Creative Commons, MIT, GNU, or Free-BSD licenses and are available for academic research.

Table 4: Overview of the language sets, which are originally obtained from CommonVoice (CV; Ardila et al., 2020), FLEURS, Googlei18n open-source project (GOP; Sodimana et al., 2018; Kjartansson et al., 2020; He et al., 2020), Living Audio dataset (LAD; Braude et al., 2019), M-AILABS Speech Dataset (MSD; Solak, 2019), NCHLT Speech Corpus (NCHLT; Barnard et al., 2014), and VoxForge (VF; MacLean, 2018).

| SET # | LANGUAGES (ISO CODE, CORPUS) |
|---|---|
| 1 | CZECH (CES, CV), MANDARIN (CMN, FLEURS) MIN NAN (NAN, CV), POLISH (POL, MSD) ROMANIAN (RON, FLEURS), SPANISH (SPA, VF) |
| 2 | CANTONESE (YUE, FLEURS), CROATIAN (HRV, FLEURS) ENGLISH (ENG, LAD), ITALIAN (ITA, FLEURS) PERSIAN (FAS, CV), SLOVAK (SLK, FLEURS) |
| 3 | KHMER (KHM, FLEURS), KOREAN (KOR, FLEURS) NORTHERN KURDISH (KMR, CV), NYNORSK (NNO, CV) SOUTHERN NDEBELE (NBL, NCHLT), TATAR (TAT, CV) |
| 4 | SINDHI (SND, FLEURS), SLOVENIAN (SLV, CV) SOUTHERN SOTHO (SOT, GOP), SPANISH (SPA, MSD) URDU (URD, FLEURS), WESTERN MARI (MRJ, CV) |
| 5 | ENGLISH (ENG, VF), GERMAN (DEU, VF) HEBREW (HEB, FLEURS), JAPANESE (JPN, FLEURS) RUSSIAN (RUS, FLEURS), SPANISH (SPA, FLEURS) |

Table 5: Dataset statistics for the training set of each of the language sets used in our experiments, in the balanced data setting. ISO codes are used for the languages, duration is presented in seconds, and transcript length is in number of characters. Averages and standard deviations are reported.

| SET # | ISO | NUMBER OF DATA POINTS | DURATION | TRANSCRIPT LENGTH | SET # | ISO | NUMBER OF DATA POINTS | DURATION | TRANSCRIPT LENGTH |
|---|---|---|---|---|---|---|---|---|---|
| 1 | CES | 908 | $4.0 \pm 1.7$ | $23.8 \pm 22.1$ | 2 | ENG | 647 | $4.7 \pm 1.5$ | $63.7 \pm 25.4$ |
| | CMN | 322 | $10.4 \pm 3.5$ | $36.8 \pm 13.9$ | | FAS | 693 | $5.2 \pm 1.7$ | $34.4 \pm 18.2$ |
| | NAN | 1406 | $2.6 \pm 0.7$ | $3.4 \pm 1.9$ | | HRV | 291 | $11.7 \pm 3.3$ | $116.3 \pm 35.7$ |
| | POL | 482 | $7.5 \pm 3.0$ | $104.6 \pm 46.3$ | | ITA | 326 | $10.7 \pm 3.2$ | $140.4 \pm 42.3$ |
| | RON | 274 | $12.6 \pm 3.1$ | $136.1 \pm 45.1$ | | SLK | 330 | $10.6 \pm 3.3$ | $116.2 \pm 38.6$ |
| | SPA | 445 | $8.1 \pm 2.2$ | $91.1 \pm 26.4$ | | YUE | 243 | $12.2 \pm 3.7$ | $31.7 \pm 10.2$ |
| 3 | KHM | 206 | $13.7 \pm 3.4$ | $122.5 \pm 36.5$ | 4 | MRJ | 707 | $5.1 \pm 2.0$ | $40.8 \pm 22.8$ |
| | KMR | 723 | $5.0 \pm 1.6$ | $30.8 \pm 15.0$ | | SLV | 918 | $3.9 \pm 1.1$ | $30.2 \pm 12.3$ |
| | KOR | 269 | $12.5 \pm 3.0$ | $45.8 \pm 14.1$ | | SND | 263 | $12.0 \pm 3.6$ | $105.4 \pm 31.2$ |
| | NBL | 744 | $4.8 \pm 1.9$ | $31.3 \pm 10.0$ | | SOT | 655 | $5.5 \pm 2.0$ | $51.0 \pm 23.6$ |
| | NNO | 709 | $4.5 \pm 1.2$ | $41.2 \pm 17.3$ | | SPA | 550 | $6.6 \pm 3.4$ | $87.2 \pm 50.2$ |
| | TAT | 835 | $4.3 \pm 1.8$ | $33.2 \pm 20.8$ | | URD | 299 | $11.3 \pm 3.4$ | $119.9 \pm 37.1$ |
| 5 | DEU | 745 | $4.8 \pm 1.6$ | $43.3 \pm 16.1$ | | | | | |
| | ENG | 712 | $5.0 \pm 1.5$ | $47.7 \pm 17.4$ | | | | | |
| | HEB | 345 | $10.2 \pm 3.3$ | $91.9 \pm 29.8$ | | | | | |
| | JPN | 290 | $11.5 \pm 3.1$ | $50.0 \pm 15.8$ | | | | | |
| | RUS | 318 | $10.8 \pm 3.4$ | $125.6 \pm 42.2$ | | | | | |
| | SPA | 311 | $11.1 \pm 3.4$ | $144.6 \pm 50.0$ | | | | | |

Table 6: Overview of the additional corpora available for the first two sets, which are originally obtained from CV, Fleurs, LAD, Multilingual Librispeech (MLS; Pratap et al., 2020), MSD, NCHLT, Spoken Wikipedia corpus (SWC; Baumann et al., 2019), VF, and Voxpopuli (VP; Wang et al., 2021).

| Set # | Language | ISO code | Corpus |
|---|---|---|---|
| 1 | Czech | CES | CV, Fleurs, VP |
| | Mandarin | CMN | CV, Fleurs |
| | Min Nan | NAN | CV |
| | Polish | POL | CV, Fleurs, MSD, MLS, VP |
| | Romanian | RON | CV, Fleurs, VP |
| | Spanish | SPA | CV, Fleurs, MSD, MLS, VF, VP |
| 2 | Cantonese | YUE | CV, Fleurs |
| | Croatian | HRV | Fleurs, VP |
| | English | ENG | CV, Fleurs, LAD, MSD, MLS, NCHLT, SWC, VF, VP |
| | Italian | ITA | CV, Fleurs, MSD, MLS, VF, VP |
| | Persian | FAS | CV, Fleurs |
| | Slovak | SLK | CV, Fleurs, VP |

## E EVALUATION METRIC DETAILS

In Section 4, we discuss the evaluation metrics used. Here, we provide more details about the computation of the CER. The CER can be computed by comparing the system generated and reference transcripts using the formula:

$$\text{CER} = \frac{I + S + D}{N} \times 100, \tag{18}$$

where $I$ is the number of insertions, $S$ the number of substitutions, and $D$ the number of deletions in a minimum edit distance alignment between the reference and system output, and $N$ is the number of characters in the reference transcript. The WER is computed identically, but operates at the word level rather than the character level (see WER results in Appendix F.2).

## F RESULTS

In Section F.1, we present the language-specific results on the development set, showing the effect of our tested hyperparameters. In addition, we show the language-specific test results in Section F.2. In this section, we include a WER analysis for set 4 for completeness. This set was chosen, as it contains languages with clear word boundaries. Additionally, we present the language-specific results of our ablation study and an analysis of the batch duration hyperparameter in Section F.3. We present multi-seed experiments on sets 1 and 3 in Section F.4. Finally, we address the effect on the best-performing language in Section F.5 and plot the group weights for additional language sets and models in Section F.6.

### F.1 LANGUAGE-SPECIFIC DEVELOPMENT RESULTS

To show the effect of our tested hyperparameters on the performance of the `CTC-DRO` models, we present language-specific results on the development set. In Table 7, we show the development results for tested values of $\eta_q \in \{10^{-3}, 10^{-4}\}$ and $\alpha \in \{0.1, 0.5, 1\}$ in the balanced data setup. The results for models trained with additional training data are shown in Table 8. For each language set, the model with the best-performing hyperparameter setting is evaluated on the test data. All results are obtained using a learning rate of $10^{-4}$.

Table 7: Results of the `CTC-DRO` models on the development set for the different language sets, where languages are indicated by their ISO code. We show the CER on the individual languages and CER averaged across languages (Avg) for fine-tuned `MMS` and `XLS-R` models. We highlight the best hyperparameter setting per set.

| Set # | Model | $\eta_q$ | $\alpha$ | CES ($\downarrow$) | CMN ($\downarrow$) | NAN ($\downarrow$) | POL ($\downarrow$) | RON ($\downarrow$) | SPA ($\downarrow$) | Avg ($\downarrow$) |
|---|---|---|---|---|---|---|---|---|---|---|
| 1 | MMS | $10^{-3}$ | 0.1 | 11.6 | 45.5 | 58.7 | 4.2 | 16.0 | 2.6 | 23.1 |
| | | $10^{-3}$ | 0.5 | 12.4 | 44.4 | 56.4 | 4.4 | 16.9 | 3.0 | 22.9 |
| | | $10^{-3}$ | 1.0 | 12.2 | 45.8 | 56.1 | 4.5 | 16.6 | 2.7 | 23.0 |
| | | $10^{-4}$ | 0.1 | 9.8 | 49.1 | 62.2 | 4.2 | 16.4 | 2.5 | 24.0 |
| | | $10^{-4}$ | 0.5 | 13.0 | 47.7 | 61.8 | 4.3 | 17.4 | 2.8 | 24.5 |
| | | **$10^{-4}$** | **1.0** | 11.7 | 45.5 | 55.2 | 4.2 | 18.0 | 2.7 | 22.9 |
| | XLS-R | $10^{-3}$ | 0.1 | 13.7 | 47.9 | 57.1 | 3.8 | 13.7 | 2.4 | 23.1 |
| | | $10^{-3}$ | 0.5 | 13.0 | 50.7 | 56.2 | 3.8 | 14.6 | 2.7 | 23.5 |
| | | $10^{-3}$ | 1.0 | 12.6 | 45.4 | 58.0 | 3.9 | 14.8 | 2.6 | 22.9 |
| | | **$10^{-4}$** | **0.1** | 12.2 | 50.7 | 55.8 | 3.6 | 14.5 | 2.4 | 23.2 |
| | | $10^{-4}$ | 0.5 | 12.2 | 50.3 | 58.9 | 3.7 | 14.9 | 2.5 | 23.8 |
| | | $10^{-4}$ | 1.0 | 13.5 | 49.2 | 58.0 | 3.7 | 15.2 | 2.8 | 23.7 |

| Set # | Model | $\eta_q$ | $\alpha$ | ENG ($\downarrow$) | FAS ($\downarrow$) | HRV ($\downarrow$) | ITA ($\downarrow$) | SLK ($\downarrow$) | YUE ($\downarrow$) | Avg ($\downarrow$) |
|---|---|---|---|---|---|---|---|---|---|---|
| 2 | MMS | $10^{-3}$ | 0.1 | 0.4 | 28.8 | 8.1 | 5.0 | 8.7 | 48.5 | 16.6 |
| | | **$10^{-3}$** | **0.5** | 0.6 | 30.0 | 8.0 | 5.2 | 7.8 | 44.8 | 16.0 |
| | | $10^{-3}$ | 1.0 | 1.1 | 28.0 | 7.9 | 5.0 | 8.9 | 45.2 | 16.0 |
| | | $10^{-4}$ | 0.1 | 0.4 | 27.3 | 7.1 | 5.0 | 7.4 | 46.8 | 15.6 |
| | | $10^{-4}$ | 0.5 | 0.9 | 27.6 | 7.6 | 4.7 | 7.7 | 45.7 | 15.7 |
| | | $10^{-4}$ | 1.0 | 1.4 | 27.9 | 8.2 | 5.9 | 8.9 | 46.5 | 16.5 |
| | XLS-R | $10^{-3}$ | 0.1 | 0.4 | 29.0 | 8.2 | 4.3 | 7.9 | 50.5 | 16.7 |
| | | $10^{-3}$ | 0.5 | 0.9 | 29.1 | 11.1 | 4.7 | 8.4 | 52.1 | 17.7 |
| | | $10^{-3}$ | 1.0 | 1.4 | 27.7 | 12.0 | 4.3 | 8.9 | 49.4 | 17.3 |
| | | $10^{-4}$ | 0.1 | 0.4 | 27.6 | 8.8 | 3.8 | 8.0 | 94.1 | 23.8 |
| | | **$10^{-4}$** | **0.5** | 0.6 | 31.6 | 9.8 | 4.3 | 8.3 | 45.2 | 16.6 |
| | | $10^{-4}$ | 1.0 | 0.8 | 31.2 | 10.7 | 4.5 | 9.5 | 47.1 | 17.3 |

| Set # | Model | $\eta_q$ | $\alpha$ | KHM ($\downarrow$) | KMR ($\downarrow$) | KOR ($\downarrow$) | NBL ($\downarrow$) | NNO ($\downarrow$) | TAT ($\downarrow$) | Avg ($\downarrow$) |
|---|---|---|---|---|---|---|---|---|---|---|
| 3 | MMS | $10^{-3}$ | 0.1 | 38.7 | 13.4 | 27.3 | 7.3 | 1.4 | 17.5 | 17.5 |
| | | $10^{-3}$ | 0.5 | 37.5 | 14.9 | 27.1 | 8.0 | 2.2 | 19.3 | 18.2 |
| | | $10^{-3}$ | 1.0 | 34.9 | 13.0 | 24.9 | 7.9 | 0.6 | 19.2 | 16.7 |
| | | **$10^{-4}$** | **0.1** | 34.3 | 12.6 | 25.5 | 7.4 | 0.8 | 16.8 | 16.2 |
| | | $10^{-4}$ | 0.5 | 35.3 | 13.8 | 26.6 | 8.3 | 0.8 | 20.8 | 17.6 |
| | | $10^{-4}$ | 1.0 | 36.8 | 13.5 | 26.4 | 7.9 | 0.5 | 20.1 | 17.5 |
| | XLS-R | $10^{-3}$ | 0.1 | 34.5 | 15.2 | 25.8 | 8.5 | 0.7 | 17.2 | 17.0 |
| | | $10^{-3}$ | 0.5 | 47.0 | 17.7 | 29.3 | 10.7 | 3.0 | 19.8 | 21.2 |
| | | $10^{-3}$ | 1.0 | 40.6 | 18.2 | 27.4 | 10.0 | 1.1 | 19.9 | 19.5 |
| | | **$10^{-4}$** | **0.1** | 33.1 | 14.9 | 29.9 | 9.3 | 2.8 | 19.5 | 18.2 |
| | | $10^{-4}$ | 0.5 | 43.6 | 16.4 | 27.8 | 9.4 | 1.1 | 22.7 | 20.2 |
| | | $10^{-4}$ | 1.0 | 46.0 | 19.6 | 28.3 | 10.7 | 2.3 | 23.5 | 21.7 |

| Set # | Model | $\eta_q$ | $\alpha$ | MRJ ($\downarrow$) | SLV ($\downarrow$) | SND ($\downarrow$) | SOT ($\downarrow$) | SPA ($\downarrow$) | URD ($\downarrow$) | Avg ($\downarrow$) |
|---|---|---|---|---|---|---|---|---|---|---|
| 4 | MMS | $10^{-3}$ | 0.1 | 9.3 | 10.3 | 19.5 | 12.5 | 4.9 | 35.4 | 15.3 |
| | | **$10^{-3}$** | **0.5** | 9.2 | 11.5 | 18.8 | 13.4 | 4.6 | 23.6 | 13.5 |
| | | $10^{-3}$ | 1.0 | 9.4 | 12.9 | 20.3 | 14.7 | 5.4 | 30.4 | 15.5 |
| | | $10^{-4}$ | 0.1 | 8.4 | 9.3 | 21.4 | 11.8 | 4.7 | 27.0 | 13.8 |
| | | $10^{-4}$ | 0.5 | 8.9 | 10.2 | 19.0 | 13.1 | 4.4 | 33.6 | 14.9 |
| | | $10^{-4}$ | 1.0 | 9.6 | 12.5 | 18.7 | 13.6 | 4.4 | 27.4 | 14.4 |
| | XLS-R | **$10^{-3}$** | **0.1** | 11.8 | 8.7 | 21.8 | 14.8 | 5.4 | 28.6 | 15.2 |
| | | $10^{-3}$ | 0.5 | 11.8 | 13.0 | 21.0 | 16.1 | 4.8 | 39.0 | 17.6 |
| | | $10^{-3}$ | 1.0 | 13.9 | 18.1 | 22.7 | 17.2 | 4.5 | 39.6 | 19.3 |
| | | $10^{-4}$ | 0.1 | 12.3 | 9.3 | 21.9 | 14.2 | 4.6 | 34.9 | 16.2 |
| | | $10^{-4}$ | 0.5 | 14.5 | 13.9 | 23.7 | 17.5 | 5.5 | 40.7 | 19.3 |
| | | $10^{-4}$ | 1.0 | 12.8 | 13.2 | 20.8 | 15.0 | 4.4 | 30.4 | 16.1 |

| Set # | Model | $\eta_q$ | $\alpha$ | DEU ($\downarrow$) | ENG ($\downarrow$) | HEB ($\downarrow$) | JPN ($\downarrow$) | RUS ($\downarrow$) | SPA ($\downarrow$) | Avg ($\downarrow$) |
|---|---|---|---|---|---|---|---|---|---|---|
| 5 | MMS | $10^{-3}$ | 0.1 | 8.3 | 13.4 | 43.5 | 54.7 | 13.3 | 8.0 | 23.5 |
| | | $10^{-3}$ | 0.5 | 10.0 | 14.1 | 31.9 | 53.0 | 13.9 | 8.7 | 21.9 |
| | | **$10^{-3}$** | **1.0** | 12.2 | 15.6 | 41.9 | 52.4 | 14.3 | 9.8 | 24.4 |
| | | $10^{-4}$ | 0.1 | 8.2 | 14.8 | 32.9 | 64.2 | 14.1 | 8.7 | 23.8 |
| | | $10^{-4}$ | 0.5 | 9.8 | 15.3 | 39.0 | 65.6 | 14.6 | 9.4 | 25.6 |
| | | $10^{-4}$ | 1.0 | 12.6 | 16.8 | 38.0 | 74.5 | 14.9 | 12.8 | 28.3 |
| | XLS-R | $10^{-3}$ | 0.1 | 7.7 | 13.0 | 40.6 | 111.5 | 12.4 | 7.7 | 32.1 |
| | | $10^{-3}$ | 0.5 | 9.2 | 13.8 | 48.8 | 119.3 | 12.9 | 28.1 | 38.7 |
| | | $10^{-3}$ | 1.0 | 11.1 | 15.5 | 48.9 | 127.7 | 16.1 | 18.2 | 39.6 |
| | | **$10^{-4}$** | **0.1** | 6.1 | 11.2 | 41.5 | 77.1 | 11.1 | 8.9 | 26.0 |
| | | $10^{-4}$ | 0.5 | 9.6 | 13.0 | 45.4 | 105.5 | 11.9 | 8.3 | 32.3 |
| | | $10^{-4}$ | 1.0 | 10.9 | 14.1 | 44.9 | 118.8 | 12.3 | 9.0 | 35.0 |

Table 8: Results of the `CTC-DRO` models on the development set for the first two language sets using additional amounts of training data per language, where languages are indicated by their ISO code. We show the CER on the individual languages and CER averaged across languages (Avg) for fine-tuned `MMS` and `XLS-R` models. We highlight the best hyperparameter setting per set.

| Set # | Model | $\eta_q$ | $\alpha$ | CES ($\downarrow$) | CMN ($\downarrow$) | NAN ($\downarrow$) | POL ($\downarrow$) | RON ($\downarrow$) | SPA ($\downarrow$) | Avg ($\downarrow$) |
|---|---|---|---|---|---|---|---|---|---|---|
| 1 | MMS | $10^{-3}$ | 0.1 | 8.4 | 57.6 | 68.6 | 6.9 | 9.5 | 5.3 | 26.1 |
| | | $10^{-3}$ | 0.5 | 8.0 | 48.6 | 64.8 | 7.0 | 9.6 | 5.4 | 23.9 |
| | | $10^{-3}$ | 1.0 | 8.5 | 50.8 | 71.5 | 7.5 | 9.8 | 5.2 | 25.5 |
| | | $10^{-4}$ | 0.1 | 8.1 | 50.1 | 64.0 | 6.6 | 9.7 | 5.2 | 24.0 |
| | | **$10^{-4}$** | **0.5** | 7.9 | 45.6 | 60.3 | 6.8 | 9.8 | 5.2 | 22.6 |
| | | $10^{-4}$ | 1.0 | 8.0 | 49.1 | 68.5 | 7.0 | 9.5 | 5.3 | 24.6 |
| | XLS-R | $10^{-3}$ | 0.1 | 9.1 | 57.8 | 67.5 | 8.1 | 11.2 | 6.6 | 26.7 |
| | | $10^{-3}$ | 0.5 | 12.9 | 57.8 | 69.8 | 10.4 | 13.2 | 7.8 | 28.7 |
| | | $10^{-3}$ | 1.0 | 11.1 | 53.3 | 67.2 | 9.3 | 12.7 | 7.5 | 26.9 |
| | | $10^{-4}$ | 0.1 | 10.6 | 61.4 | 70.1 | 9.3 | 11.6 | 6.9 | 28.3 |
| | | $10^{-4}$ | 0.5 | 12.7 | 56.7 | 69.7 | 10.0 | 13.5 | 8.0 | 28.4 |
| | | **$10^{-4}$** | **1.0** | 12.3 | 52.9 | 67.2 | 10.3 | 13.7 | 8.3 | 27.5 |

| Set # | Model | $\eta_q$ | $\alpha$ | ENG ($\downarrow$) | FAS ($\downarrow$) | HRV ($\downarrow$) | ITA ($\downarrow$) | SLK ($\downarrow$) | YUE ($\downarrow$) | Avg ($\downarrow$) |
|---|---|---|---|---|---|---|---|---|---|---|
| 2 | MMS | $10^{-3}$ | 0.1 | 9.4 | 20.5 | 8.1 | 7.2 | 10.8 | 53.4 | 18.2 |
| | | $10^{-3}$ | 0.5 | 9.6 | 20.4 | 8.8 | 7.5 | 11.3 | 52.4 | 18.3 |
| | | $10^{-3}$ | 1.0 | 9.5 | 19.5 | 8.9 | 7.5 | 10.8 | 49.8 | 17.6 |
| | | $10^{-4}$ | 0.1 | 9.6 | 18.8 | 8.6 | 7.5 | 10.5 | 55.1 | 18.4 |
| | | $10^{-4}$ | 0.5 | 9.4 | 20.3 | 8.4 | 7.5 | 10.9 | 48.2 | 17.5 |
| | | **$10^{-4}$** | **1.0** | 9.4 | 19.9 | 8.9 | 7.4 | 11.3 | 47.8 | 17.5 |
| | XLS-R | $10^{-3}$ | 0.1 | 11.6 | 24.6 | 10.2 | 9.0 | 13.4 | 56.9 | 21.0 |
| | | $10^{-3}$ | 0.5 | 11.7 | 22.7 | 9.7 | 8.2 | 12.9 | 57.9 | 20.5 |
| | | $10^{-3}$ | 1.0 | 23.2 | 30.7 | 18.4 | 15.3 | 21.7 | 83.0 | 32.1 |
| | | $10^{-4}$ | 0.1 | 11.5 | 25.7 | 10.1 | 8.0 | 12.8 | 91.0 | 26.5 |
| | | $10^{-4}$ | 0.5 | 19.2 | 27.0 | 16.3 | 12.6 | 18.9 | 68.6 | 27.1 |
| | | **$10^{-4}$** | **1.0** | 11.6 | 25.1 | 9.6 | 9.1 | 14.4 | 50.3 | 20.0 |

## F.2 LANGUAGE-SPECIFIC TEST RESULTS

For each language set, we present the language-specific test set results of our experiments using balanced training data in Table 9. Table 10 shows the language-specific test set results for the first two sets based on experiments using all available training data in `ML-SUPERB 2.0`. In Table 11, we present results using WER on set 4 (balanced setup for brevity), which contains languages with clear word boundaries. Using this evaluation metric, `CTC-DRO` still achieves substantial worst-language improvements, namely 22.3% (`MMS`) and 11.8% (`XLS-R`) relative WER reductions. For `MMS`, the average WER is substantially reduced (14.4% relative). For `XLS-R`, the average WER increased marginally (0.4% relative), even though the average CER improved. This shows that character-level and word-level improvements do not always align, as a single character error invalidates an entire word. This also causes different languages to emerge as worst-performing under the CER versus the WER metrics. Despite the slight average WER increase for one model, `CTC-DRO` achieves its primary objective of substantially improving the performance on the worst-performing language.

Table 9: Results of the baseline models (`Base`), `group DRO` models (`GDRO`), and `CTC-DRO` models (`Ours`) on the test set for the different language sets, where languages are indicated by their ISO code. We show the CER on the individual languages, CER averaged across languages (`Avg CER`), and LID accuracy (`LID`) for fine-tuned `MMS` and `XLS-R` models. Best LID and CER results are highlighted, and the CERs for the worst-performing languages are underlined.

| SET # | MODEL | TYPE | CES (↓) | CMN (↓) | NAN (↓) | POL (↓) | RON (↓) | SPA (↓) | AVG CER (↓) | LID (↑) |
|---|---|---|---|---|---|---|---|---|---|---|
| 1 | MMS | BASE | 8.4 | 52.4 | 60.8 | 3.6 | 13.3 | 1.8 | 23.4 | **97.4** |
| | | GDRO | 20.6 | 48.6 | 86.6 | 4.3 | 16.7 | 6.2 | 30.5 | 78.7 |
| | | OURS | 10.5 | 46.1 | 56.8 | 3.7 | 17.9 | 2.3 | **22.9** | 95.8 |
| | XLS-R | BASE | 7.3 | 64.9 | 60.8 | 3.1 | 13.4 | 1.8 | 25.2 | **92.6** |
| | | GDRO | 27.4 | 48.9 | 78.4 | 3.7 | 14.9 | 6.6 | 30.0 | 87.8 |
| | | OURS | 7.8 | 50.7 | 57.6 | 3.0 | 14.2 | 1.8 | **22.5** | 89.5 |

| SET # | MODEL | TYPE | ENG (↓) | FAS (↓) | HRV (↓) | ITA (↓) | SLK (↓) | YUE (↓) | AVG CER (↓) | LID (↑) |
|---|---|---|---|---|---|---|---|---|---|---|
| 2 | MMS | BASE | 0.2 | 21.8 | 9.0 | 5.9 | 8.2 | 49.4 | 15.8 | **98.4** |
| | | GDRO | 11.8 | 29.7 | 10.8 | 6.2 | 10.2 | 55.5 | 20.7 | 98.2 |
| | | OURS | 0.5 | 22.1 | 8.8 | 5.5 | 8.6 | 44.4 | **15.0** | 96.2 |
| | XLS-R | BASE | 0.1 | 20.6 | 10.9 | 4.6 | 8.9 | 68.8 | 19.0 | **94.2** |
| | | GDRO | 12.7 | 28.5 | 14.4 | 5.1 | 10.2 | 58.8 | 21.6 | 87.0 |
| | | OURS | 0.5 | 21.5 | 12.6 | 5.2 | 10.0 | 45.0 | **15.8** | 89.3 |

| SET # | MODEL | TYPE | KHM (↓) | KMR (↓) | KOR (↓) | NBL (↓) | NNO (↓) | TAT (↓) | AVG CER (↓) | LID (↑) |
|---|---|---|---|---|---|---|---|---|---|---|
| 3 | MMS | BASE | 31.3 | 12.2 | 34.2 | 7.4 | 2.5 | 9.0 | 16.1 | 98.5 |
| | | GDRO | 33.2 | 19.1 | 34.0 | 22.4 | 9.8 | 13.5 | 22.0 | **98.7** |
| | | OURS | 31.3 | 12.0 | 27.6 | 8.1 | 2.3 | 10.2 | **15.3** | **98.7** |
| | XLS-R | BASE | 33.2 | 13.3 | 32.3 | 8.7 | 3.7 | 11.0 | **17.0** | **99.2** |
| | | GDRO | 38.0 | 23.9 | 35.5 | 26.6 | 11.9 | 14.9 | 25.1 | 97.2 |
| | | OURS | 32.2 | 14.8 | 31.9 | 10.1 | 5.0 | 12.0 | 17.7 | 97.9 |

| SET # | MODEL | TYPE | MRJ (↓) | SLV (↓) | SND (↓) | SOT (↓) | SPA (↓) | URD (↓) | AVG CER (↓) | LID (↑) |
|---|---|---|---|---|---|---|---|---|---|---|
| 4 | MMS | BASE | 14.8 | 6.9 | 24.0 | 14.4 | 5.9 | 20.1 | 14.4 | 87.9 |
| | | GDRO | 13.1 | 14.4 | 19.0 | 17.1 | 3.8 | 21.8 | 14.9 | **91.9** |
| | | OURS | 17.7 | 8.1 | 17.5 | 11.4 | 4.4 | 18.4 | **12.9** | 87.3 |
| | XLS-R | BASE | 14.0 | 4.8 | 23.3 | 11.6 | 4.2 | 29.7 | 14.6 | 88.4 |
| | | GDRO | 19.5 | 25.6 | 18.5 | 23.0 | 3.9 | 21.1 | 18.6 | 83.5 |
| | | OURS | 11.9 | 6.7 | 21.0 | 13.8 | 4.8 | 24.2 | **13.7** | **88.9** |

| SET # | MODEL | TYPE | DEU (↓) | ENG (↓) | HEB (↓) | JPN (↓) | RUS (↓) | SPA (↓) | AVG CER (↓) | LID (↑) |
|---|---|---|---|---|---|---|---|---|---|---|
| 5 | MMS | BASE | 5.4 | 11.1 | 30.2 | 90.0 | 12.0 | 7.2 | 26.0 | **96.3** |
| | | GDRO | 27.6 | 27.0 | 32.6 | 62.2 | 17.6 | 8.4 | 29.2 | 67.0 |
| | | OURS | 10.9 | 15.4 | 39.2 | 57.5 | 13.2 | 9.3 | **24.3** | 90.5 |
| | XLS-R | BASE | 4.8 | 9.2 | 33.2 | 114.8 | 10.5 | 7.1 | 29.9 | 89.0 |
| | | GDRO | 29.1 | 26.8 | 46.1 | 92.9 | 16.5 | 9.3 | 36.8 | 57.7 |
| | | OURS | 5.7 | 9.6 | 38.6 | 71.5 | 10.1 | 7.3 | **23.8** | **91.0** |

Table 10: Results of the baseline models (`Base`), `group DRO` models (`GDRO`), and `CTC-DRO` models (`Ours`) on the test set for the first two language sets using additional amounts of training data per language, where languages are indicated by their ISO code. We show the CER on the individual languages, CER averaged across languages (`Avg CER`), and LID accuracy (`LID`) for fine-tuned `MMS` and `XLS-R` models. Best LID and CER results are highlighted, and the CERs for the worst-performing languages are underlined.

| SET # | MODEL | TYPE | CES (↓) | CMN (↓) | NAN (↓) | POL (↓) | RON (↓) | SPA (↓) | AVG CER (↓) | LID (↑) |
|---|---|---|---|---|---|---|---|---|---|---|
| 1 | MMS | BASE | 9.1 | 58.9 | 67.5 | 6.0 | 7.1 | 5.0 | 25.6 | 98.1 |
| | | GDRO | 13.8 | 92.1 | 96.3 | 6.7 | 11.9 | 5.8 | 37.8 | 83.9 |
| | | OURS | 8.7 | 45.9 | 62.8 | 6.2 | 7.5 | 5.3 | **22.8** | **98.5** |
| | XLS-R | BASE | 13.0 | 92.1 | 78.3 | 9.8 | 12.0 | 8.5 | 35.6 | 96.4 |
| | | GDRO | 18.9 | 86.4 | 90.8 | 5.7 | 21.6 | 5.0 | 38.1 | 72.3 |
| | | OURS | 12.9 | 52.5 | 67.5 | 9.0 | 11.9 | 7.8 | **26.9** | **97.1** |

| | MODEL | TYPE | ENG (↓) | FAS (↓) | HRV (↓) | ITA (↓) | SLK (↓) | YUE (↓) | AVG CER (↓) | LID (↑) |
|---|---|---|---|---|---|---|---|---|---|---|
| 2 | MMS | BASE | 9.6 | 16.9 | 8.5 | 6.8 | 8.0 | 66.9 | 19.5 | 99.0 |
| | | GDRO | 10.1 | 70.0 | 24.3 | 7.9 | 14.8 | 105.4 | 38.8 | 81.0 |
| | | OURS | 9.7 | 18.1 | 8.3 | 6.6 | 7.3 | 48.1 | **16.4** | **99.1** |
| | XLS-R | BASE | 11.9 | 32.2 | 9.6 | 8.1 | 9.2 | 97.2 | 28.0 | 98.2 |
| | | GDRO | 8.8 | 88.2 | 33.9 | 6.7 | 23.3 | 102.9 | 44.0 | 80.8 |
| | | OURS | 11.6 | 23.2 | 9.3 | 8.2 | 8.9 | 51.4 | **18.8** | **98.6** |

Table 11: Results of the baseline models (`Base`), `group DRO` models (`GDRO`), and `CTC-DRO` models (`Ours`) on the test set for set 4, where languages are indicated by their ISO code. We show the WER on the individual languages, WER averaged across languages (`Avg WER`), and LID accuracy (`LID`) for fine-tuned `MMS` and `XLS-R` models. Best LID and WER results are highlighted, and the WERs for the worst-performing languages are underlined.

| SET # | MODEL | TYPE | MRJ (↓) | SLV (↓) | SND (↓) | SOT (↓) | SPA (↓) | URD (↓) | AVG WER (↓) | LID (↑) |
|---|---|---|---|---|---|---|---|---|---|---|
| 4 | MMS | BASE | 59.2 | 32.4 | 65.9 | 52.1 | 30.1 | 56.4 | 49.4 | 87.9 |
| | | GDRO | 57.3 | 56.1 | 50.3 | 61.5 | 56.6 | 50.3 | 50.2 | **91.9** |
| | | OURS | 51.2 | 36.7 | 49.4 | 43.6 | 22.5 | 50.3 | **42.3** | 87.3 |
| | XLS-R | BASE | 60.2 | 22.9 | 63.9 | 44.6 | 21.4 | 74.0 | **47.8** | 88.4 |
| | | GDRO | 71.3 | 82.5 | 51.0 | 75.8 | 19.8 | 57.2 | 59.6 | 83.5 |
| | | OURS | 58.8 | 29.2 | 59.5 | 51.0 | 24.1 | 65.3 | 48.0 | **88.9** |

## F.3 ABLATION STUDY

We present the language-specific results of our ablation study in Table 12.

Table 12: Results of the baseline models (`Base`) and `CTC-DRO` models (`Ours`) on the test set for set 5 with ablations removing the length-matched group losses (`Dur`) and smoothed maximization objective (`Smooth`). We show the CER averaged across languages (`Avg CER`) as well as the CER on the individual languages and the LID accuracy (`LID`) for fine-tuned `MMS` and `XLS-R` models. Best LID and CER results are highlighted, and the CERs for the worst-performing languages are underlined.

| MODEL | TYPE | DEU (↓) | ENG (↓) | HEB (↓) | JPN (↓) | RUS (↓) | SPA (↓) | AVG CER (↓) | LID (↑) |
|---|---|---|---|---|---|---|---|---|---|
| MMS | BASE | 5.4 | 11.1 | 30.2 | 90.0 | 12.0 | 7.2 | 26.0 | **96.3** |
| | OURS | 10.9 | 15.4 | 39.2 | 57.5 | 13.2 | 9.3 | **24.3** | 90.5 |
| | - DUR | 19.4 | 21.2 | 30.9 | 84.6 | 12.9 | 8.3 | 29.6 | 66.1 |
| | - SMOOTH | 95.6 | 96.0 | 98.8 | 102.1 | 97.4 | 97.3 | 97.9 | 13.2 |
| XLS-R | BASE | 4.8 | 9.2 | 33.2 | 114.8 | 10.5 | 7.1 | 29.9 | 89.0 |
| | OURS | 5.7 | 9.6 | 38.6 | 71.5 | 10.1 | 7.3 | **23.8** | **91.0** |
| | - DUR | 35.6 | 36.5 | 72.9 | 115.2 | 27.4 | 15.9 | 50.6 | 54.4 |
| | - SMOOTH | 18.5 | 24.5 | 69.9 | 194.2 | 41.2 | 19.9 | 61.4 | 43.2 |

To assess the sensitivity of our results to the choice of the batch duration hyperparameter (Algorithm 1), we first report in Table 13 the total audio duration per batch used in our main experiments for each language set. We then perform an additional robustness experiment on language set 5 by training additional baseline and CTC-DRO models with half the duration target. Table 14 shows the test set performance. With the smaller duration target, CTC-DRO achieves relative worst-language CER reductions of 34.2% for MMS and 15.8% for XLS-R compared to the corresponding baselines. With the original batch duration target of roughly 50 seconds, the relative reductions are 36.1% and 37.7%, respectively. While XLS-R shows more sensitivity to the choice of the duration target, both models maintain substantial improvements from CTC-DRO across both settings.

Table 13: Batch duration statistics for the main experiments. For each language set, we report the maximum total audio duration in seconds.

| SET # | TOTAL AUDIO DURATION / BATCH (S) |
|---|---|
| 1 | 50.2 |
| 2 | 48.6 |
| 3 | 54.8 |
| 4 | 47.9 |
| 5 | 46.0 |

Table 14: Effect of the batch duration hyperparameter on the test set for set 5. We show the CER of the worst-performing language (Max CER, ISO code for the worst-performing language provided as ISO) as well as the average CER (Avg CER) and LID accuracy (LID) for baseline (Base) and CTC-DRO models (Ours). Best results are highlighted.

| SET # | MODEL | DURATION (S) (AUDIO / BATCH) | TYPE | MAX CER (ISO) (↓) | AVG CER (↓) | LID (↑) |
|---|---|---|---|---|---|---|
| 5 | MMS | 23.0 | BASE | 79.5 (JPN) | 24.9 | **92.7** |
| | MMS | 23.0 | OURS | **52.3** (JPN) | **21.7** | 90.8 |
| | XLS-R | 23.0 | BASE | 101.9 (JPN) | **29.0** | **86.6** |
| | XLS-R | 23.0 | OURS | **85.8** (JPN) | 30.3 | 77.8 |
| 5 | MMS | 46.0 | BASE | 90.0 (JPN) | 26.0 | **96.3** |
| | MMS | 46.0 | OURS | **57.5** (JPN) | **24.3** | 90.5 |
| | XLS-R | 46.0 | BASE | 114.8 (JPN) | 29.9 | 89.0 |
| | XLS-R | 46.0 | OURS | **71.5** (JPN) | **23.8** | **91.0** |

## F.4 ROBUSTNESS EXPERIMENTS

To assess the robustness of our results across random seeds, we perform experiments on language sets 1 and 3 using four unique random seeds and report the results in Table 15. We selected these sets, because they showed the smallest single-seed improvements of CTC-DRO compared to the baseline (see Table 1). Overall, the largest gains in worst-language CER are stable across seeds. For the remaining language sets, where the single-seed gaps between CTC-DRO and the baseline are substantially larger, we expect the conclusions to be at least as robust.

Table 15: Results of the baseline models (Base) and CTC-DRO models (Ours) on the test sets for set 1 and 3 using four random seeds. We show the mean and standard deviation of the worst-language CER (Max CER) as well as the mean difference in worst-language CER (Avg Delta) between the baseline (Base) and CTC-DRO models (Ours) for fine-tuned MMS and XLS-R models.

| SET # | MODEL | BASE MAX CER (MEAN ± SD) | OURS MAX CER (MEAN ± SD) | AVG DELTA (BASE − OURS) | SEEDS WITH LOWER CER (OUT OF 4) |
|---|---|---|---|---|---|
| 1 | MMS | 58.7 ± 2.1 | 56.6 ± 1.1 | 2.1 | 3 |
| | XLS-R | 76.4 ± 15.0 | 58.6 ± 2.5 | 17.9 | 4 |
| 3 | MMS | 32.0 ± 1.6 | 31.3 ± 0.7 | 0.7 | 2 |
| | XLS-R | 33.2 ± 1.0 | 34.7 ± 2.2 | −1.5 | 2 |

### F.5 BEST-PERFORMING LANGUAGE RESULTS

To directly address the effect on the best-performing language groups, we investigate the CER of the best-performing language per setup (i.e., lowest CER reported in Table 9 and 10) and average scores across sets and models. In the balanced data setup, the average CER of the best-performing language is 3.0% (standard deviation (SD) 2.1%) for the baseline, 3.7% (SD 2.7%) for CTC-DRO, and 6.6% (SD 3.0%) for group DRO. A paired t-test shows no statistically significant difference between the baseline and CTC-DRO ($p = 0.19$), while there is a significant difference between the baseline and group DRO ($p = 0.0068$), with group DRO having worse performance (6.6% vs. 3.0%). In the unbalanced data setup, the average CER of the best-performing language is 7.1% (SD 1.6%) for the baseline, 7.0% (SD 1.3%) for CTC-DRO, and 6.4% (SD 1.2%) for group DRO. Paired t-tests show no significant difference between the baseline and CTC-DRO ($p = 0.61$) or between the baseline and group DRO ($p = 0.53$). Thus, CTC-DRO does not significantly degrade best-language performance, while achieving substantial worst-language improvements.

### F.6 COMPARISON OF GROUP WEIGHTS

In Section 6.2, we analyze the behavior of group DRO and CTC-DRO models during training for XLS-R on set 5. Here, we include additional visualizations, showing the behavior of MMS models on sets 5 and 2 in Figures 3 and 4, respectively. These visualizations confirm that the stability pattern extends to different models and language sets, showing that group DRO exhibits substantial weight fluctuations, while CTC-DRO maintains more stable group weights throughout training.

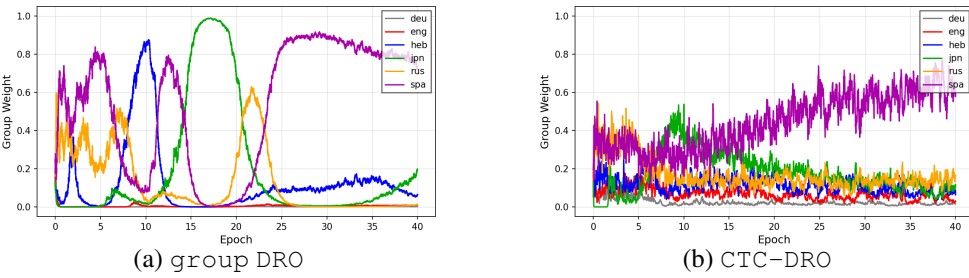

(a) group DRO  (b) CTC-DRO

Figure 3: Group weights for each language throughout training of an MMS model trained with group DRO or CTC-DRO on balanced data from set 5.

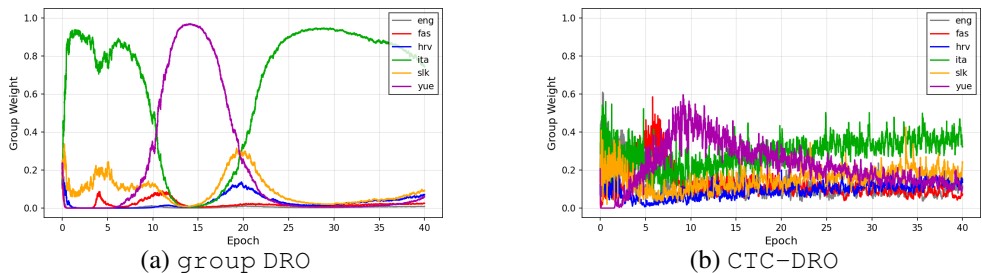

(a) group DRO  (b) CTC-DRO

Figure 4: Group weights for each language throughout training of an MMS model trained with group DRO or CTC-DRO on balanced data from set 2.

## G NORMALIZATION EXPERIMENTS

We conduct additional experiments to explain why normalization of the CTC loss alone is insufficient (see Section 2.3). We evaluate four approaches on language set 1 (balanced setup): (1) group DRO with losses normalized by the number of frames in the sequence (FRAME); (2) group DRO with losses normalized by the number of target labels (TARGET); (3) CTC-DRO without our new batch sampler that computes length-matched group losses (instead using the group DRO batch sampler) and with losses normalized by the number of frames in the sequence (FRAME; NO LENGTH-MATCHED); (4) CTC-DRO without our new batch sampler that computes length-matched group

losses (instead using the `group DRO` batch sampler) and with losses normalized by the number of target labels (TARGET; NO LENGTH-MATCHED). These experiments follow the same experimental setup used for our main experiments.

Normalizing each utterance's loss by its own length (number of input frames or target labels) also scales the corresponding gradient. The longest utterances are most strongly downweighted, while the gradients of shorter utterances retain relatively more weight within a batch. Importantly, longer sequences inherently provide more information and should influence the gradients more, so reducing their gradients limits the model's ability to learn from the most informative examples. We note that a different global learning rate would not compensate for this per-utterance imbalance. We present the test set results of this experiment in Table 16 and confirm that simple normalization provides no solution to address the problem of incomparable CTC losses across languages.

Table 16: CER of the worst-performing language (`Max CER`, ISO code for the worst-performing language provided as `ISO`), as well as the average CER (`Avg CER`) and LID accuracy (`LID`) across languages for the baseline models (`Base`), `group DRO` models (GDRO), and `CTC-DRO` models (`Ours`) on the test set for set 1 under different normalization settings. We also report the step size $\eta_q$ and smoothing $\alpha$ selected on the development set where applicable. Best results are highlighted.

| SET # | MODEL | TYPE | $\eta_q$ | $\alpha$ | MAX CER (ISO) ($\downarrow$) | AVG CER ($\downarrow$) | LID ($\uparrow$) |
|---|---|---|---|---|---|---|---|
| 1 | MMS | BASE (NONE) | – | – | **60.8** (NAN) | **23.4** | 97.4 |
| | | GDRO (NONE) | $10^{-4}$ | – | 86.6 (NAN) | 30.5 | 78.7 |
| | | GDRO (FRAME) | $10^{-4}$ | – | 91.5 (CMN) | 32.8 | **98.1** |
| | | GDRO (TARGET) | $10^{-4}$ | – | 170.7 (CMN) | 87.0 | 65.4 |
| | | OURS (FRAME; NO LENGTH-MATCHED) | $10^{-4}$ | 0.5 | 94.7 (CMN) | 31.9 | 97.9 |
| | | OURS (TARGET; NO LENGTH-MATCHED) | $10^{-4}$ | 0.1 | 98.7 (CMN) | 43.7 | 83.6 |
| | XLS-R | BASE (NONE) | – | – | **64.9** (CMN) | **25.2** | 92.6 |
| | | GDRO (NONE) | $10^{-4}$ | – | 78.4 (NAN) | 30.0 | 87.8 |
| | | GDRO (FRAME) | $10^{-3}$ | – | 81.2 (CMN) | 33.2 | **94.2** |
| | | GDRO (TARGET) | $10^{-3}$ | – | 119.9 (CMN) | 95.0 | 44.3 |
| | | OURS (FRAME; NO LENGTH-MATCHED) | $10^{-3}$ | 0.5 | 67.6 (CMN) | 26.6 | 93.7 |
| | | OURS (TARGET; NO LENGTH-MATCHED) | $10^{-4}$ | 0.1 | 119.7 (CMN) | 50.2 | 78.7 |

# H  SCALABILITY EXPERIMENTS

The strong performance of `CTC-DRO` motivates investigating the algorithm's scalability. While our algorithm adds minimal computational costs, a rigorous hyperparameter search for any new, large-scale experiment is inherently resource-intensive (our main experiments already required training 130 models over approximately 1500 GPU hours). To validate scalability under our compute budget, we conducted a single, challenging scaling experiment on a diverse set of 18 languages, extending the languages in set 1 by 12 randomly selected languages. This appendix shows the full experiment, presenting the language-corpus pairs (Section H.1), the development set results from our hyperparameter search (Section H.2), and the final test set performance (Section H.3).

## H.1  DATASETS

Table 17 shows the language-corpus pairs that are included in our scaling experiments for the balanced setup and when additional training data is available.

## H.2  LANGUAGE-SPECIFIC DEVELOPMENT RESULTS

Tables 18 and 19 show the language-specific performance on the development set from our hyperparameter search. We tested values of $\eta_q \in \{10^{-3}, 10^{-4}\}$ and $\alpha \in \{0.1, 0.5, 1\}$, while keeping the learning rate fixed at $10^{-4}$. Table 18 shows the results for the balanced data setup, while Table 19 contains the results for models trained with additional training data. From this evaluation, the best-performing hyperparameter setting was selected for evaluation on the test data.

Table 17: Overview of the languages included in the scaling experiment, which are originally obtained from CV, Fleurs, LAD, MLS, MSD, NCHLT, SWC, VF, and VP.

| SETUP | LANGUAGES (ISO CODE, CORPORA) |
|---|---|
| BALANCED | BASHKORT (BAK, CV), BURMESE (MYA, FLEURS) |
| | MANDARIN (CMN, CV), MIN NAN (NAN, CV) |
| | CANTONESE (YUE, CV), CZECH (CES, CV) |
| | ENGLISH (ENG, LAD), FRENCH (FRA, MLS) |
| | GERMAN (DEU, VF), GUARANI (GRN, CV) |
| | ITALIAN (ITA, FLEURS), KHMER (KHM, FLEURS) |
| | PERSIAN (FAS, CV), POLISH (POL, MSD) |
| | ROMANIAN (RON, FLEURS), RUSSIAN (RUS, LAD) |
| | SPANISH (SPA, VF), SWATI (SSW, NCHLT) |
| ADDITIONAL DATA | BASHKORT (BAK, CV), BURMESE (MYA, FLEURS) |
| | CANTONESE (YUE, CV, FLEURS), MANDARIN (CMN, CV, FLEURS) |
| | MIN NAN (NAN, CV), CZECH (CES, CV, FLEURS, VP) |
| | ENGLISH (ENG, CV, FLEURS, LAD, MSD, MLS, NCHLT, SWC, VF, VP), |
| | FRENCH (FRA, CV, FLEURS, MSD, MLS, VF, VP), |
| | GERMAN (DEU, CV, FLEURS, MSD, MLS, SWC, VF, VP), GUARANI (GRN, CV) |
| | ITALIAN (ITA, CV, FLEURS, MSD, VF, VP), KHMER (KHM, FLEURS) |
| | PERSIAN (FAS, CV, FLEURS), POLISH (POL, CV, FLEURS, MSD, MLS, VP) |
| | ROMANIAN (RON, CV, FLEURS, VP), RUSSIAN (RUS, CV, FLEURS, LAD, MSD, VF) |
| | SPANISH (SPA, CV, FLEURS, MSD, MLS, VF, VP), SWATI (SSW, NCHLT) |

Table 18: Results of the `CTC-DRO` models on the development set, where languages are indicated by their ISO code. We show the CER on the individual languages and CER averaged across languages (`Avg CER`) for fine-tuned `MMS` and `XLS-R` models. We highlight the best hyperparameter setting per set.

| LANGUAGE | MMS | | | | | | XLS-R | | | | | |
|---|---|---|---|---|---|---|---|---|---|---|---|---|
| $\eta_q$ | $10^{-3}$ | | | $10^{-4}$ | | | $10^{-3}$ | | | $10^{-4}$ | | |
| $\alpha$ | 0.1 | **0.5** | 1.0 | 0.1 | 0.5 | 1.0 | 0.1 | 0.5 | **1.0** | 0.1 | 0.5 | 1.0 |
| BAK ($\downarrow$) | 20.7 | 11.9 | 12.7 | 10.6 | 11.6 | 12.8 | 21.6 | 39.5 | 33.1 | 35.3 | 31.9 | 32.8 |
| CES ($\downarrow$) | 24.0 | 13.2 | 15.5 | 11.6 | 14.4 | 16.6 | 23.9 | 45.7 | 41.1 | 40.4 | 39.9 | 34.9 |
| CMN ($\downarrow$) | 74.7 | 54.6 | 55.1 | 57.1 | 57.9 | 57.9 | 78.0 | 86.4 | 84.1 | 90.2 | 75.4 | 65.4 |
| DEU ($\downarrow$) | 14.5 | 8.7 | 9.8 | 7.7 | 10.0 | 11.7 | 13.6 | 31.2 | 27.6 | 28.6 | 27.6 | 26.3 |
| ENG ($\downarrow$) | 6.6 | 0.8 | 1.5 | 1.4 | 2.1 | 2.8 | 5.2 | 8.7 | 8.4 | 7.0 | 5.1 | 3.0 |
| FAS ($\downarrow$) | 43.5 | 31.6 | 33.0 | 32.7 | 32.4 | 33.9 | 38.6 | 57.9 | 52.3 | 54.3 | 53.1 | 54.4 |
| FRA ($\downarrow$) | 29.4 | 19.9 | 20.2 | 18.5 | 18.5 | 18.6 | 23.2 | 45.8 | 43.8 | 45.3 | 43.0 | 43.7 |
| GRN ($\downarrow$) | 19.4 | 12.1 | 14.6 | 10.0 | 13.5 | 15.0 | 21.3 | 40.5 | 33.8 | 33.4 | 32.1 | 36.0 |
| ITA ($\downarrow$) | 13.8 | 5.5 | 6.8 | 5.7 | 6.0 | 6.4 | 13.6 | 33.1 | 28.3 | 27.7 | 27.1 | 26.1 |
| KHM ($\downarrow$) | 76.6 | 39.0 | 41.6 | 36.5 | 36.4 | 38.6 | 87.4 | 78.2 | 85.8 | 91.9 | 77.5 | 80.9 |
| MYA ($\downarrow$) | 74.1 | 35.2 | 31.0 | 28.7 | 30.2 | 30.3 | 54.4 | 90.1 | 89.3 | 74.5 | 89.6 | 88.2 |
| NAN ($\downarrow$) | 77.9 | 56.4 | 63.2 | 63.9 | 66.8 | 72.1 | 75.3 | 80.4 | 83.5 | 80.7 | 81.4 | 77.5 |
| POL ($\downarrow$) | 10.0 | 4.8 | 5.3 | 4.8 | 4.5 | 5.0 | 7.7 | 20.9 | 17.8 | 18.6 | 18.1 | 18.5 |
| RON ($\downarrow$) | 28.9 | 17.3 | 17.9 | 17.8 | 17.6 | 16.2 | 23.8 | 47.4 | 40.6 | 44.7 | 43.5 | 43.1 |
| RUS ($\downarrow$) | 14.4 | 1.3 | 2.5 | 3.1 | 3.3 | 4.0 | 12.3 | 18.1 | 14.5 | 16.8 | 6.5 | 2.8 |
| SPA ($\downarrow$) | 8.6 | 3.5 | 4.5 | 3.7 | 5.0 | 5.6 | 8.8 | 28.2 | 23.4 | 23.9 | 23.7 | 22.4 |
| SSW ($\downarrow$) | 15.3 | 9.1 | 13.1 | 6.6 | 12.1 | 15.3 | 16.4 | 32.0 | 29.6 | 26.8 | 29.4 | 22.7 |
| YUE ($\downarrow$) | 61.7 | 41.2 | 42.6 | 43.2 | 44.5 | 49.3 | 66.3 | 82.0 | 77.8 | 82.5 | 69.4 | 57.9 |
| AVG CER ($\downarrow$) | 34.1 | 20.3 | 21.7 | 20.2 | 21.5 | 22.9 | 32.9 | 48.1 | 45.3 | 45.7 | 43.0 | 40.9 |

## H.3 LANGUAGE-SPECIFIC TEST RESULTS

Table 20 summarizes test set performance for all languages in the balanced setup and Table 21 shows results when models are trained on all available `ML-SUPERB 2.0` data. We find that `CTC-DRO` maintains its effectiveness at improving the performance on the worst-performing language at a larger scale. On the balanced data setup, `CTC-DRO` reduces worst-language CER by 8.9% relative for `MMS` and 9.2% relative for `XLS-R`. For `XLS-R`, the average CER improves by 17.2% relative. While `MMS` shows a slight average CER increase (3.0% relative), it successfully reduces the worst-language performance, which is our primary objective. On the unbalanced data setup, `XLS-R` shows

Table 19: Results of the `CTC-DRO` models on the development set using additional amounts of training data per language, where languages are indicated by their ISO code. We show the CER on the individual languages and CER averaged across languages (`Avg`) for fine-tuned `MMS` and `XLS-R` models. We highlight the best hyperparameter setting per set.

| LANGUAGE | MMS | | | | | | XLS-R | | | | | |
|---|---|---|---|---|---|---|---|---|---|---|---|---|
| $\eta_q$ | $10^{-3}$ | | | $10^{-4}$ | | | $10^{-3}$ | | | $10^{-4}$ | | |
| $\alpha$ | 0.1 | **0.5** | 1.0 | 0.1 | 0.5 | 1.0 | 0.1 | 0.5 | **1.0** | 0.1 | 0.5 | 1.0 |
| BAK (↓) | 87.9 | 14.2 | 19.4 | 12.0 | 13.3 | 14.5 | 61.7 | 16.6 | 19.4 | 13.8 | 19.0 | 18.3 |
| CES (↓) | 84.6 | 9.2 | 11.4 | 9.0 | 9.2 | 9.2 | 45.7 | 10.2 | 11.4 | 8.7 | 10.3 | 11.6 |
| CMN (↓) | 247.1 | 48.2 | 54.9 | 55.7 | 48.8 | 47.6 | 103.6 | 56.9 | 54.9 | 51.6 | 53.2 | 47.4 |
| DEU (↓) | 70.6 | 9.2 | 10.4 | 9.1 | 8.9 | 9.3 | 37.0 | 9.8 | 10.4 | 9.4 | 9.5 | 10.3 |
| ENG (↓) | 73.3 | 10.9 | 12.2 | 10.6 | 10.6 | 10.7 | 44.3 | 11.5 | 12.2 | 10.5 | 10.9 | 11.1 |
| FAS (↓) | 98.7 | 22.4 | 23.8 | 23.8 | 23.7 | 23.4 | 65.9 | 23.3 | 23.8 | 22.3 | 25.0 | 24.4 |
| FRA (↓) | 70.5 | 12.3 | 14.1 | 12.4 | 12.4 | 12.3 | 44.2 | 13.0 | 14.1 | 11.9 | 12.5 | 13.0 |
| GRN (↓) | 88.6 | 14.3 | 24.1 | 11.4 | 15.9 | 16.9 | 54.9 | 20.9 | 24.1 | 15.7 | 21.3 | 22.0 |
| ITA (↓) | 77.2 | 8.9 | 9.4 | 8.6 | 8.8 | 8.2 | 31.7 | 8.5 | 9.4 | 7.9 | 8.7 | 8.8 |
| KHM (↓) | 99.9 | 31.4 | 39.7 | 32.5 | 30.0 | 30.0 | 90.2 | 38.6 | 39.7 | 37.4 | 34.9 | 36.2 |
| MYA (↓) | 94.7 | 28.1 | 47.1 | 29.5 | 32.0 | 28.3 | 89.3 | 65.4 | 47.1 | 74.3 | 30.4 | 29.6 |
| NAN (↓) | 163.7 | 67.7 | 70.5 | 69.2 | 70.4 | 69.4 | 99.8 | 70.7 | 70.5 | 62.6 | 71.7 | 71.3 |
| POL (↓) | 78.3 | 8.5 | 8.6 | 7.9 | 7.8 | 8.3 | 37.0 | 7.6 | 8.6 | 7.9 | 7.9 | 9.2 |
| RON (↓) | 74.6 | 10.3 | 12.3 | 10.4 | 10.8 | 11.2 | 42.2 | 12.1 | 12.3 | 11.6 | 11.2 | 11.8 |
| RUS (↓) | 90.2 | 9.9 | 12.7 | 9.8 | 9.9 | 10.0 | 46.2 | 11.6 | 12.7 | 9.5 | 11.7 | 12.1 |
| SPA (↓) | 75.8 | 5.9 | 6.5 | 6.0 | 5.7 | 6.0 | 30.9 | 5.6 | 6.5 | 5.5 | 6.0 | 6.7 |
| SSW (↓) | 97.0 | 14.3 | 23.6 | 11.8 | 16.2 | 16.1 | 49.1 | 24.4 | 23.6 | 12.1 | 20.5 | 18.4 |
| YUE (↓) | 261.2 | 50.4 | 55.7 | 53.3 | 50.7 | 50.1 | 96.0 | 55.6 | 55.7 | 45.4 | 51.6 | 46.3 |
| **AVG CER (↓)** | 107.4 | 20.9 | 25.4 | 21.3 | 21.4 | 21.2 | 59.4 | 25.7 | 25.4 | 23.2 | 23.1 | 22.7 |

Table 20: Results of the baseline models and `CTC-DRO` models on the test set, where languages are indicated by their ISO code. We show the CER on the individual languages, CER averaged across languages (`Avg CER`), and LID accuracy (`LID`) for fine-tuned `MMS` and `XLS-R` models. Best LID and CER results are highlighted, and the CERs for the worst-performing languages are underlined.

| | MMS | | XLS-R | |
|---|---|---|---|---|
| LANGUAGE | BASELINE | CTC-DRO | BASELINE | CTC-DRO |
| BAK (↓) | 12.6 | 14.9 | 30.4 | 23.7 |
| CES (↓) | 10.3 | 13.4 | 28.8 | 22.2 |
| CMN (↓) | 65.2 | 55.6 | 94.9 | 78.8 |
| DEU (↓) | 5.6 | 8.2 | 22.0 | 13.9 |
| ENG (↓) | 0.8 | 0.8 | 2.7 | 5.2 |
| FAS (↓) | 23.5 | 25.2 | 45.4 | 34.0 |
| FRA (↓) | 14.4 | 16.5 | 37.0 | 20.6 |
| GRN (↓) | 6.4 | 11.0 | 31.8 | 21.9 |
| ITA (↓) | 5.5 | 6.9 | 24.3 | 12.4 |
| KHM (↓) | 34.0 | 33.8 | 67.7 | 86.2 |
| MYA (↓) | 35.3 | 40.6 | 91.6 | 61.6 |
| NAN (↓) | 66.1 | 60.2 | 81.7 | 78.4 |
| POL (↓) | 3.7 | 4.3 | 15.6 | 7.3 |
| RON (↓) | 14.0 | 16.9 | 38.0 | 23.3 |
| RUS (↓) | 5.1 | 1.6 | 7.3 | 12.0 |
| SPA (↓) | 2.1 | 3.2 | 13.4 | 7.5 |
| SSW (↓) | 6.3 | 13.1 | 14.3 | 19.1 |
| YUE (↓) | 47.5 | 43.2 | 74.7 | 69.2 |
| **AVG CER (↓)** | **19.9** | 20.5 | 40.1 | **33.2** |
| **LID (↑)** | **96.5** | 94.7 | 84.0 | **84.9** |

particularly strong results, namely a reduction of 23.7% relative for the worst-performing language and 9.9% relative average CER improvement. For `MMS`, `CTC-DRO` still reduces the worst-language CER (although marginally), while maintaining comparable average performance.

Table 21: Results of the baseline models and `CTC-DRO` models on the test set using additional amounts of training data per language, where languages are indicated by their ISO code. We show the CER on the individual languages, CER averaged across languages (`Avg CER`), and LID accuracy (`LID`) for fine-tuned `MMS` and `XLS-R` models. Best LID and CER results are highlighted, and the CERs for the worst-performing languages are underlined.

| | MMS | | XLS-R | |
| --- | --- | --- | --- | --- |
| LANGUAGE | BASELINE | CTC-DRO | BASELINE | CTC-DRO |
| BAK ($\downarrow$) | 13.0 | 14.3 | 14.3 | 21.6 |
| CES ($\downarrow$) | 8.6 | 10.3 | 8.6 | 11.3 |
| CMN ($\downarrow$) | 60.7 | 48.1 | 75.7 | 56.3 |
| DEU ($\downarrow$) | 8.8 | 9.5 | 8.4 | 10.2 |
| ENG ($\downarrow$) | 9.4 | 10.7 | 9.3 | 12.3 |
| FAS ($\downarrow$) | 18.1 | 18.5 | 17.1 | 22.2 |
| FRA ($\downarrow$) | 12.9 | 13.4 | 12.5 | 14.8 |
| GRN ($\downarrow$) | 6.7 | 12.8 | 9.4 | 21.1 |
| ITA ($\downarrow$) | 7.8 | 7.7 | 6.8 | 8.6 |
| KHM ($\downarrow$) | 37.1 | 32.0 | 68.5 | 40.7 |
| MYA ($\downarrow$) | 30.8 | 28.6 | 95.5 | 44.1 |
| NAN ($\downarrow$) | 70.6 | 70.0 | 75.3 | 72.9 |
| POL ($\downarrow$) | 6.2 | 6.9 | 6.3 | 7.9 |
| RON ($\downarrow$) | 7.5 | 8.7 | 7.7 | 10.5 |
| RUS ($\downarrow$) | 9.4 | 9.7 | 8.7 | 12.7 |
| SPA ($\downarrow$) | 5.1 | 5.6 | 5.1 | 6.3 |
| SSW ($\downarrow$) | 5.5 | 16.6 | 7.5 | 26.4 |
| YUE ($\downarrow$) | 53.0 | 51.3 | 70.8 | 56.7 |
| AVG CER ($\downarrow$) | **20.6** | 20.8 | 28.2 | **25.4** |
| LID ($\uparrow$) | **97.6** | **97.6** | 96.2 | 95.2 |

## I  TRAINING TIMES

In Table 22, we present averaged wall-clock training times for baseline and `CTC-DRO` models across our main experiments. Each model was trained on a single NVIDIA RTX A6000 GPU.

Table 22: Averaged wall-clock training times for baseline and `CTC-DRO` models across experiments using balanced and additional training data in seconds.

| SET # | BASELINE TIME (S) | CTC-DRO TIME (S) |
| --- | --- | --- |
| 1-5 (BALANCED DATA) | 24,665 | 24,986 |
| 1-2 (ADDITIONAL DATA) | 81,122 | 82,458 |

