# OpenReview forum: "CTC-DRO: Robust Optimization for Reducing Language Disparities in Speech Recognition"
_ICLR.cc/2026/Conference — ICLR 2026 Poster_

### Official Review · Reviewer_id6Z · 2025-10-21

**Soundness:** 3
**Presentation:** 2
**Contribution:** 3
**Rating:** 6
**Confidence:** 3

**Summary:**

This paper proposes an algorithm for ASR training on datasets composed of multiple groups such as languages. The contributions are two fold: the authors enforce balanced minibatches with equal total sample length per group (by audio duration), and they introduce a novel update rule that reweights updates according to each group’s loss contribution. An ablation study shows that both components are necessary. In multilingual settings, the method yields substantial improvements on the worst-performing languages and on the overall average compared with the baseline and group DRO. Performance on language identification is more mixed.

**Strengths:**

The paper has several strengths:

-  Results: consistent improvements on the worst-performing languages and on the overall average versus both the baseline and group DRO.

- Practicality: the method is lightweight, adds minimal complexity, and does not introduce notable computational overhead; useful for real usage.

- The experiments are concise with a proper ablation study showing that both contribution matter, plus complementary scaling experiments on group size.

**Weaknesses:**

Despite the overall good results, the paper has some weaknesses :

- The results on (language identification) LID are mixed; significance tests would help determine whether the outcomes are due to randomness or indicate genuine advantages of the baseline, group DRO, or CTC-DRO.

- The effect of the procedure on the best-performing language groups is not discussed. Appendix results appear to suggest baseline ≥ CTC-DRO ≥ group DRO. This issue matters in practice and should be analyzed explicitly. I don’t view this as a major flaw (CTC-DRO often outperforms group DRO) but the results are somewhat mixed, and a statistical analysis would help clarify the picture.

- The choice of Equation (10) is not clearly justified and seems motivated by the form of the resulting update. A clearer rationale for this specific form would help readers build intuition and extend the method; otherwise, it may be cleaner to start directly from the group DRO objective (Equation (7)) and propose the extension explicitly using the already existing rationale in the paper.

**Questions:**

On Equation (14), the justification for "proportionality" is, to me, unsatisfactory. The Lagrange multipliers lambda and lambda_g are intermediate quantities used to reach the solution and will depend on L_g. Without making their dependence on L_g explicit, you can’t claim that q_g is proportional.

If we push the computation further: starting from (14), q_g = L_g / (lambda + lambda_g) − alpha, with the constraints sum_g q_g = 1 and sum_g (lambda_g q_g) = 0 (KKT). For the active set A = { g : q_g > 0 }, complementary slackness gives lambda_g = 0. Hence, for g in A, q_g = L_g / lambda − alpha. Summing over g in A, 1 = sum_{g in A} q_g = (sum_{g in A} L_g) / lambda − |A| * alpha, which implies lambda = (sum_{g in A} L_g) / (1 + |A| * alpha). Therefore, for g in A, q_g = L_g / lambda − alpha = [ L_g * (1 + |A| * alpha) / sum_{g in A} L_g ] − alpha. Thus, the relationship is not simple proportionality in L_g; but in L_g /sum_g' L_g'. This doesn’t necessarily weaken the update choice rationale, but the claim should be stated precisely. I may have made a mistake or missed something and my derivation should be checked.

- The actual chosen per-group batch duration (target total audio length) is never stated. What value did you use, and how sensitive are the results to this choice?

- Have you tried the procedure with different group definitions (e.g., dialect, SNR bins, speaker gender/age, domain, or per-language subsets)? If so, how do the results vary?

---

> ### Author Response · Authors · 2025-11-24
> **Author response**
>
> We thank the reviewer for the careful and constructive review. Below we address the reviewer’s concerns and questions.
>
> **Language identification**
>
> We appreciate the reviewer’s point about LID performance. We would like to clarify that we report LID results for completeness (and note that LID is generally high for both baseline and CTC-DRO models), and that CER (and in particular worst-language CER) is our primary evaluation metric. The joint ASR and LID prediction task is the standard setup for ML-SUPERB 2.0, which we follow in the paper (see lines 263–273). This joint formulation is also standard practice in multilingual ASR more broadly (e.g., in the Whisper multilingual ASR model [1]). We will make this clearer in the revised version of our paper.
>
> [1] Alec Radford, Jong Wook Kim, Tao Xu, Greg Brockman, Christine Mcleavey, and Ilya Sutskever.
> Robust Speech Recognition via Large-ScaleWeak Supervision. In Proceedings of the International
> Conference on Machine Learning, 2023.
>
> **Best-performing languages**
>
> To directly address the effect on the best-performing language groups, we looked at the CER of the best-performing language per setup (i.e., lowest CER) using the results already reported in the paper and averaged scores across sets and models. In the balanced data setup, the average CER of the best-performing language is 3.0% (SD 2.1%) for the baseline, 3.7% (SD 2.7%) for CTC-DRO, and 6.6% (SD 3.0%) for group DRO. A paired t-test shows no statistically significant difference between baseline and CTC-DRO (p = 0.19), while there is a significant difference between baseline and group DRO (p = 0.0068), with group DRO having worse performance (6.6% vs 3.0%). In the unbalanced data setup, the average CER of the best-performing language is 7.1% (SD 1.6%) for the baseline, 7.0% (SD 1.3%) for CTC-DRO, and 6.4% (SD 1.2%) for group DRO. Paired t-tests show no significant difference between baseline and CTC-DRO (p = 0.61) or between baseline and group DRO (p = 0.53). Thus, CTC-DRO does not significantly degrade best-language performance while achieving the substantial worst-language improvements reported in the paper. We will add this analysis to the revision.
>
> **Choice of Equation 10**
>
> We thank the reviewer for this comment. Our goal was to smooth the group weights so that they still increase with group losses but avoid the weight collapse and instability we observe with group DRO. In practice, we first specified a smoothed group-weight update with these properties (Equation 11) and then constructed Equation 10 as an equivalent objective whose associated weight-update rule reproduces this behavior. In the revised version, we will add a brief explanation about Equation 10 that clarifies that the objective was chosen precisely to realize the smoothed group-weight dynamics we describe.
>
> **Proportionality in Equation 14**
>
> We appreciate the reviewer catching this and would like to thank the reviewer for the detailed derivation. We agree that our original wording was imprecise. We will update the text in the revised version and adopt the more precise formulation suggested by the reviewer.
>
> **Batch duration**
>
> We thank the reviewer for this question. The details of our batch-duration choice and robustness are described in our response to Reviewer UcsN (Hyperparameter robustness). In brief, we set the duration hyperparameter d so that a batch fits in the memory of a single NVIDIA A6000 GPU and keep this value fixed for all methods on a given encoder and language set (resulting in roughly 50 seconds of audio per batch). As noted in that response, we also trained baseline and CTC-DRO models on one language set with half this duration target (about 25 seconds) and still observed reductions in worst-language CER for CTC-DRO. In the revised version, we will make the values of d explicit and include the results of this new experiment in an appendix table.
>
> **Scope of group definitions**
>
> We appreciate the reviewer's question about other group definitions. We focus on the setting of multilingual ASR and have not applied CTC-DRO to other types of groups such as dialect, SNR, gender/age, or domain. While CTC-DRO could in principle be applied to these alternative groupings, we view them as out of scope for this paper but an interesting direction for future work, which we will mention in the revision.
>
> We hope this addresses the reviewer's concerns and thank them for their thoughtful feedback.

---

### Official Review · Reviewer_LdZ2 · 2025-10-30

**Soundness:** 2
**Presentation:** 3
**Contribution:** 3
**Rating:** 4
**Confidence:** 4

**Summary:**

This paper proposes CTC-DRO, a robust optimization method for multilingual ASR under the CTC framework. Standard group DRO fails because CTC losses scale with input length and differ across languages, leading to unstable weight updates. CTC-DRO mitigates this by using length-matched batching so group losses are comparable, and introducing a smoothed maximization objective for group weight updates.

**Strengths:**

1. The paper addresses the language disparities in the multilingual ASR, which is critical.
2. The proposed modifications are simple, well-motivated, and easily adoptable in existing pipelines.
3. Results show consistent and sometimes substantial improvements over both CTC and Group DRO baselines, especially for worst-case languages.

**Weaknesses:**

1. Experiments focus on small subsets (5–6 languages per set), with only limited scaling experiments. It is unclear how well the method generalizes to large-scale multilingual ASR (50–100+ languages).
2. The paper does not compare with alternative ASR+LID strategies, such as auxiliary CTC objectives [A] or condition-aware SSL representations [B], which directly integrate language identification and also improve the low-resource language performance.
3. Although an ablation study is included, it is limited to one dataset. More extensive analysis across different training settings would better clarify the contribution of each component.

[A] Chen, W., Yan, B., Shi, J., Peng, Y., Maiti, S., & Watanabe, S. (2023, June). Improving massively multilingual asr with auxiliary ctc objectives. In ICASSP 2023-2023 IEEE International Conference on Acoustics, Speech and Signal Processing (ICASSP) (pp. 1-5). IEEE.

[B] Lu, Y. J., Liu, J., Thebaud, T., Moro-Velazquez, L., Rastrow, A., Dehak, N., & Villalba, J. (2024). CA-SSLR: Condition-Aware Self-Supervised Learning Representation for Generalized Speech Processing. Advances in Neural Information Processing Systems, 37, 50126-50151.

**Questions:**

1. How does CTC-DRO scale to very large multilingual ASR settings (dozens or hundreds of languages)?
2. Are the reported gains consistent across multiple random seeds?
3. Could this approach generalize to other sequence modeling losses (RNN-T, seq2seq)?
4. Could CTC-DRO be extended to code-switching scenarios, where disparities are often more severe [C]?

[C] Liu, H., Garcia, L. P., Zhang, X., Khong, A. W., & Khudanpur, S. (2024, April). Enhancing code-switching speech recognition with interactive language biases. In ICASSP 2024-2024 IEEE International Conference on Acoustics, Speech and Signal Processing (ICASSP) (pp. 10886-10890). IEEE.

---

> ### Author Response · Authors · 2025-11-24
> **Author response (part 1)**
>
> We thank the reviewer for their thoughtful review. We address the reviewer’s critiques and questions below.
>
> **Scaling experiments**
>
> We acknowledge that our experiments focus on smaller language sets (6-18 languages) rather than the 50-100+ language scale the reviewer asks about. This choice was deliberate, as we prioritized rigorous evaluation across multiple diverse language sets to thoroughly validate that our method is effective. Our experiments cover 27 unique languages total and follow the experimental setup of the original group DRO paper [1]. We note that our 6-language sets align with recent state-of-the-art multilingual ASR systems such as IBM’s Granite (5 languages) models. The 18-language experiment in Appendix H demonstrates that our method's effectiveness persists as we scale up. While we have not yet tested at 50-100+ languages due to computational constraints, we agree this is an important direction for future work.
>
> [1] Shiori Sagawa, Pang Wei Koh, Tatsunori B. Hashimoto, and Percy Liang. Distributionally Robust Neural Networks for Group Shifts: On the Importance of Regularization for Worst-Case Generalization. In Proceedings of the International Conference on Learning Representations, 2020.
>
> **Comparing to alternative approaches**
>
> We thank the reviewer for pointing out these related works.These works focus on architectural and representation-level improvements to include language information. Our work differs fundamentally in its objective, namely CTC-DRO directly targets worst-group performance. In our experiments, we already use joint ASR and LID training (Section 4) with high LID accuracy (Section 5), but substantial performance differences between languages remain for the baseline models. In principle, CTC-DRO could be combined with architectures like [A] and [B]. For example, our method could be applied to their CTC loss, while keeping their other mechanisms unchanged. We will add discussion of these complementary works to our related work section.
>
> **Ablation study**
>
> As per the reviewer’s suggestion, we conducted an additional ablation on language set 1. We present the results below and confirm that both components of CTC-DRO are essential. The consistent patterns across multiple sets validate our approach.
>
> | Model | Type     | Max CER (ISO) | Avg CER |  LID   |
> |:-----|:---------|--------------:|--------:|-------:|
> | MMS   | Baseline | 60.8 (nan)    |   23.4  |  97.4  |
> |       | CTC-DRO  | 56.8 (nan)    |   22.9  |  95.8  |
> |       | - Dur    | 65.6 (nan)   |   24.8  |  98.1 |
> |       | - Smooth | 59.7 (nan)   |   23.5  |  96.3 |
> | XLSR  | Baseline | 64.9 (cmn)    |   25.2  |  92.6  |
> |       | CTC-DRO  | 57.6 (nan)    |   22.5  |  89.5  |
> |       | - Dur    | 67.3 (nan)   |   25.4  |  93.9 |
> |       | - Smooth | 59.7 (nan)   |   22.7  |  98.1 |

---

> ### Author Response · Authors · 2025-11-24
> **Author response (part 2)**
>
> **Multiple random seeds**
>
> We thank the reviewer for raising this question. To assess robustness, we re-ran MMS and XLS-R on two language sets with the smallest improvements of CTC-DRO compared to the baseline (1 and 3) using four unique random seeds and report the worst-language CER results below. Overall, these new results suggest that the largest gains on worst-language CER are stable across seeds. We emphasize that we deliberately evaluated two language sets (1 and 3) where the single-seed worst-language improvements were smallest. For the remaining language sets, where the single-seed gaps between CTC-DRO and the baseline are substantially larger, we expect the conclusions to be at least as robust. We will add the table below to the appendix and explicitly refer to these multi-seed results in the revised version.
>
> | Set | Model | Baseline worst-language CER (mean ± sd) | CTC-DRO worst-language CER (mean ± sd) | Mean delta CER (baseline − CTC-DRO) | Seeds with lower CER under CTC-DRO (out of 4) |
> |:---|:-----|----------------------------------------:|---------------------------------------:|------------------------------------:|-----------------------------------------------:|
> | 1   | MMS   | 58.7 ± 2.1                              | 56.6 ± 1.1                             |                                 2.1 |                                          3    |
> |     | XLSR  | 76.4 ± 15.0                             | 58.6 ± 2.5                             |                                17.9 |                                          4    |
> | 3   | MMS   | 32.0 ± 1.6                              | 31.3 ± 0.7                             |                                 0.7 |                                          2    |
> |     | XLSR  | 33.2 ± 1.0                              | 34.7 ± 2.2                             |                               −1.5  |                                          2    |
>
> **Generalization to other losses**
>
> We thank the reviewer for this insightful question. While our experiments focus on CTC-based multilingual ASR, the smoothed worst-group objective we introduce is defined in terms of generic group losses and can, in principle, be applied to other sequence modeling losses (e.g., autoregressive token-level NLL). In ongoing work, we are exploring extensions of our approach to Whisper-style models. A full report of these experiments is beyond the scope of the present paper, but we will clarify this potential for generalization in the revised version.
>
> **Code-switching**
>
> Extending CTC-DRO to code-switching scenarios is promising. While this is out of scope for the current paper, we agree this would be valuable future work and will mention this in the revised version. We thank the reviewer for this suggestion.
>
> We hope that our responses address the reviewer’s concerns and thank them again for the helpful feedback.

---

### Official Review · Reviewer_sUDj · 2025-10-31

**Soundness:** 3
**Presentation:** 3
**Contribution:** 3
**Rating:** 4
**Confidence:** 3

**Summary:**

This paper focuses on the training algorithm of CTC ASR models. The authors improve upon the vanilla Group DRO algorithm to better align with CTC loss (which yields higher loss values for longer sequences and breaks the group weights assignment in Group DRO). Experiments on multilingual ASR confirm the effectiveness of the proposed algorithm.

**Strengths:**

The proposed algorithm appears to yield strong gains over baselines. Experiments are solid. The presentation is clear enough.

**Weaknesses:**

The contribution is a bit incremental over an existing algorithm (Group DRO). Also, I have some questions about the compared baselines in this work, see the Questions section below.

**Questions:**

1. In table 3, the row "- SMOOTH" shows a huge performance drop, even much worse than baseline. Why is that? Isn't "-SMOOTH" reduced to a standard CTC training with similar total durations across all training batches?
2. For the baseline, I think a more reasonable setting is to have a baseline that is a standard CTC trained with some length normalization.  The main observation/motivation of the proposed algorithm is that CTC places high loss on longer sequences, so what if you simply normalize CTC loss by sequence length?

---

> ### Author Response · Authors · 2025-11-24
> **Author response**
>
> We thank the reviewer for their comments. We respectfully highlight that our contributions address fundamental incompatibilities that make group DRO fail in this domain (as shown in our experiments). Our work contributes a new smoothed maximization objective and duration-matched batching, which are algorithmic contributions that enable robust optimization of multilingual CTC-based ASR models. We are happy to address their questions below.
>
> **Ablation study**
>
> We appreciate the reviewer’s question about the ablation study, as this question highlights a key finding. The “-SMOOTH” setting is not standard CTC training, as it still uses group DRO’s weight update mechanism. This configuration validates the necessity of our smoothed maximization objective. We will make this more clear in the revision of the paper.
>
> **Length normalization**
>
> We extensively tested this approach in Appendix G (page 20). Table 13 shows that both frame normalization and target sequence length normalization lead to substantially worse performance. An intuitive reason is that length normalization reduces gradients from longer sequences, which are more informative and should contribute more to learning. Additionally, normalization doesn't address different irreducible loss components across languages due to linguistic and acoustic properties. Our approach maintains natural gradient magnitudes while preventing any group from dominating training through duration-matched batching and smoothed weight updates.
>
> We hope this addresses the reviewer's concerns and thank them for their valuable feedback.

---

### Official Review · Reviewer_UcsN · 2025-11-03

**Soundness:** 3
**Presentation:** 3
**Contribution:** 3
**Rating:** 6
**Confidence:** 3

**Summary:**

The paper targets the problem of cross-language disparities in multilingual ASR when training with CTC. To solve that, they explore using the classical ML method of Group Distributionally Robust Optimization (Group DRO). First, they found out that the method is not effective when directly applied to the CTC loss, as this loss is not comparable across groups. CTC losses are systematically higher for reasons not aligned with downstream error (e.g., input length and irreducible loss differences), thereby hurting worst-group generalization. To fix this, they propose CTC-DRO, which introduces a smoothed maximization objective and implements length-matched batching to stabilize group weights and correct for group-wise mismatches in loss scale. Through empirical evaluation on the ML-SUPERB 2.0 benchmark across multiple language sets and state-of-the-art models (MMS, XLS-R), CTC-DRO demonstrates consistent improvements over baselines and Group DRO, reducing the worst-language error and narrowing language disparities in ASR and LID performance.

**Strengths:**

- The authors build a clear problem framing, pointing to a concrete issue in ASR, particularly between CTC loss geometry and group DRO weight updates, with a neat derivation showing why standard group-DRO weights can collapse to one group. The solution is well motivated, enough general, and theoretically principled.


- The present compelling empirical evidence on a credible benchmark (ML-SUPERB 2.0), with DRO-CTC showing consistent improvements on worst-group CER across five language sets and both XLS-R and MMS encoders.

- The work presents informative ablations and behavior analysis. The group-weight traces help clarify why it works (smoother, more balanced weights, and fewer collapses).


- The work is clearly documented, making it easy to reproduce. Authors promised to release code upon acceptance.

**Weaknesses:**

- Robustness to hyperparameter alpha: While the paper discusses theoretically and reports practical ranges for alpha, it does not include a systematic sensitivity analysis. A more systematic hyperparameter sensitivity (including ηq and the batch-duration target) and early/late-training stability plots across sets would increase confidence in robustness and tune-free usability.
- The work is limited to the scope of assuming languages as the sole group, relying on datasets where groups are well-defined and large enough. The work does not address potential unknown/latent groups, despite being briefly mentioned by the authors in the conclusion.
- Baselines: While the paper compares to Group DRO and standard CTC, it does not benchmark against alternative fairness-promoting or loss-calibrated approaches specifically designed for sequence models, despite mentioning them in the related work. The methods may include invariant objectives, variance-regularized DRO, loss calibration across groups, data reweighting, or even group-awre reinforcement learning. The inclusion or at least discussion of such alternatives for this particular case would strengthen the empirical case.

**Questions:**

- Would the smoothed update help beyond CTC, with autoregressive losses (token-level NLL) where length effects differ?
- Please consider adding a fixed-budget grid over alpha, showing worst-language CER stability, to help establish robustness of the gains to alpha.

---

> ### Author Response · Authors · 2025-11-24
> **Author response (part 1)**
>
> We thank the reviewer for the thoughtful and constructive feedback. Below we address their main concerns and questions.
>
> **Hyperparameter robustness**
>
> We agree with the reviewer about the importance of hyperparameter analysis, and want to clarify that we included hyperparameter search results in Appendix F.1 (Tables 7 and 8) and Appendix H.2 (Tables 15 and 16). These tables show a grid search over our key hyperparameters, α and ηq, and reveal that CTC-DRO benefits from hyperparameter tuning rather than being completely tune-free. The reviewer’s feedback makes it clear that we should refer more explicitly to these results. We will add a short discussion to the main text that will clearly state the best-performing hyperparameter values found in our search (from Appendix F.1).
>
> Regarding early/late-training stability plots, we would like to note that Section 6.2 includes group weight traces for CTC-DRO and group DRO on set 5 with XLS-R, and shows that CTC-DRO yields smoother, more balanced weights with fewer collapses. To strengthen this, we will add additional group weight evolution figures for other language sets and the MMS model, using the same visualization style. These new visualizations show that the qualitative stability pattern is not specific to a single model and language set. If the reviewer instead had different types of stability plots in mind, we would appreciate clarification and would be happy to include such plots in the revision.
>
> Regarding the batch duration hyperparameter d in Algorithm 1, we set d so that a batch fits within the memory of a single NVIDIA A6000 GPU (i.e., the hardware we used) and then kept that value fixed for all methods on that encoder and set. This resulted in batches with a duration of roughly 50 seconds across language sets. As a simple robustness check, we also trained baseline and CTC-DRO models on set 5 with half this duration target (i.e., ~25 seconds) and observed CER reductions on the worst language for the CTC-DRO models compared to the baseline of 34.2% relative (MMS) and 15.8% relative (XLS-R), compared to 36.1% and 37.7% relative respectively with a batch size of approximately 50 seconds. While XLS-R shows more sensitivity to batch duration, both models maintain substantial improvements across different batch sizes. We will add these results in an appendix table and make the values of d explicit.
>
> | Set | Model | Type     | Max CER (ISO) | Avg CER |  LID  |
> |:---|:-----|:---------|--------------:|--------:|------:|
> |  5  | MMS   | Baseline | 79.5 (jpn)    |   24.9  |  92.7 |
> |     |       | CTC-DRO  | 52.3 (jpn)    |   21.7  |  90.8 |
> |     | XLSR  | Baseline | 101.9 (jpn)   |   29.0  |  86.6 |
> |     |       | CTC-DRO  | 85.8 (jpn)    |   30.3  |  77.8 |
>
> **Scope of group definitions**
>
> The reviewer correctly notes our intentional focus on pre-defined groups. This matches the standard group DRO setting [1], which assumes group labels are available at training time. Our goal in this paper is to reduce disparities between languages, so taking language as the group is a natural choice given that ML-SUPERB provides reliable language labels and sufficient data per language (as do other datasets, so this is not unique to our setting). Methodologically, CTC-DRO only requires a discrete group label per example and it does not depend on how these labels are obtained. In the revised version, we will clarify that combining CTC-DRO with separate methods for discovering unknown or latent groups is a natural direction for future work.
>
> [1] Shiori Sagawa, Pang Wei Koh, Tatsunori B. Hashimoto, and Percy Liang. Distributionally Robust Neural Networks for Group Shifts: On the Importance of Regularization for Worst-Case Generalization. In Proceedings of the International Conference on Learning Representations, 2020.

---

> > ### Author Response · Authors · 2025-11-24
> > **Author response (part 2)**
> >
> > **Baselines**
> >
> > We appreciate the reviewer’s suggestions about our selection of baselines. We would like to note that our submission discusses why several fairness-promoting and loss-calibrated approaches are not directly applicable to our multilingual CTC setting (we do compare with length normalization approaches in Appendix G) and in the revision we will make this more explicit. Specifically, in the related-work section on robustness to distribution shifts, we group prior methods into domain-invariance, invariant-prediction and DRO, and write that we build on group DRO because it has outperformed other approaches in subpopulation-shift benchmarks [1], which is the setting we study. For calibration or reweighting-style methods, Section 2.3 discusses approaches that calibrate group losses or approximate disparities with simpler models [2, 3] and explains that they would either require substantially more computation or a proxy for group difficulty, for which there is no reliable model for speech, so instead of learning a separate calibration model we change how group losses are computed and used with CTC-DRO. For other DRO variants that still update on group-averaged CTC losses, the incompatibility analysis in Section 2.3 continues to apply. As long as losses are systematically misaligned with error (because of length and irreducible components), simply upweighting high-loss groups does not reliably improve the worst language. Finally, group-aware reinforcement-learning methods would require decoding during training and optimizing a sequence-level reward such as CER. This would however be much more expensive than the scalar group-weight update used by CTC-DRO. In the revision, we will add a short paragraph to Section 7, so that it is clear why the suggested baselines are not strictly necessary for supporting our claims in this problem setting.
> >
> > [1] Pang Wei Koh, Shiori Sagawa, Henrik Marklund, Sang Michael Xie, Marvin Zhang, Akshay Balsubramani, Weihua Hu, Michihiro Yasunaga, Richard Lanas Phillips, Irena Gao, Tony Lee, Etienne David, Ian Stavness, Wei Guo, Berton Earnshaw, Imran Haque, Sara M Beery, Jure Leskovec, Anshul Kundaje, Emma Pierson, Sergey Levine, Chelsea Finn, and Percy Liang. WILDS: A Benchmark of in-the-Wild Distribution Shifts. In Proceedings of the International Conference on
> > Machine Learning, 2021.
> >
> > [2] Yonatan Oren, Shiori Sagawa, Tatsunori B. Hashimoto, and Percy Liang. Distributionally robust
> > language modeling. In Proceedings of the Conference on Empirical Methods in Natural Language Processing and the International Joint Conference on Natural Language Processing, 2019.
> >
> > [3] Agnieszka Słowik and Leon Bottou. On distributionally robust optimization and data rebalancing. In Proceedings of the International Conference on Artificial Intelligence and Statistics, 2022.
> >
> > **Autoregressive losses**
> >
> > The reviewer asks whether the smoothed update could help beyond CTC. While our experiments focus on CTC-based multilingual ASR, the smoothed maximization objective we introduce only requires group-level losses, so in principle it could be applied to other sequence modeling losses (e.g., autoregressive token-level NLL). Currently, we are exploring extensions of our approach to Whisper-style models, but including these experiments in the current paper is outside our scope. We will add a brief remark to the revision to emphasize this potential broader applicability as future work.
> >
> > We hope this addresses the reviewer's concerns and would like to thank them again for their suggestions.

---

### Author Response · Authors · 2025-12-02
**General response**

We thank the reviewers for their thoughtful comments on our work. We are glad that reviewers recognized the importance of the problem studied in our work, found our solution well-motivated and theoretically principled, noted consistent and substantial performance improvements over baselines, and highlighted that our method is practical and easily adoptable. We have addressed all concerns and questions in our individual responses and updated the submission accordingly.

The core contributions and main experiments are unchanged. The minor revisions include an expanded discussion of baselines and complementary architectural methods, clarifications of the ablation study, evaluation, and theoretical analysis, additional robustness experiments (multi-seed, batch duration sensitivity, additional ablation), a statistical analysis of best-language performance, additional visualizations of group weights during training, and a discussion of broader applicability.

To the best of our knowledge, CTC-DRO is the first robust optimization method to successfully reduce cross-lingual performance disparities in ASR. Across five language sets (27 unique languages), balanced and unbalanced data setups, and two pre-trained models, CTC-DRO achieves consistent improvements with relative reductions in worst-language error by up to 47.1% and average error by up to 32.9%, without significantly degrading best-language performance.

---

### Meta-Review · Area_Chair_M7aB · 2025-12-30

**Summary:**

1. [UcsN] The paper would be stronger with a more systematic examination of the sensitivity of the proposed CTC-DRO algorithm to the various hyperparameters, especially the smoothing parameter, $\alpha$.
2. [UcsN] The paper is limited by its focus on languages as the sole grouping variable.
3. [LdZ2] The paper would be stronger if it considered a larger number of languages.
4. [UcsN] The paper would be stronger if it benchmarked against or at least discussed other fairness-promoting approaches.
5. [LdZ2] The paper should compare to alternative approaches such as auxiliary CTC objectives or condition-aware SSL representations.
6. [sUDj] Wouldn't a length-normalized CTC model be a more sensible baseline?
7. [sUDj] Why is the [-SMOOTH] condition in Table 3 so much worse?
8. [LdZ2] The ablation study is limited because it considers only one dataset.
9. [id6Z] LID results would benefit from significance testing.
10. [id6Z] The paper should state the effect of the CTC-DRO training on the best-performing language.
11. [id6Z] Equation 10 needs to be better justified.
12. [id6Z] The proportionality in Equation 14 needs to be better justified.

**Reviewer Concerns:**

1. This concern is fully addressed in the rebuttal and revision. The authors pointed out that the requested sensitivity analysis had already been done in Appendix F.1 (Tables 7 and 8) and Appendix H.2 (Tables 15 and 16), and that the results show that tuning of the hyperparameters is important for obtaining the best performance. Also, the revision adds a sentence in Section 4 emphasizing that the CRO-specific hyperparameters were tuned and pointing to Appendix F. The authors also point out that one such requested early/late stability plot is present in Section 6.2, and they added additional plots to the revised manuscript. Finally, the authors add results from a second set of experiments in which the batch duration is halved to illustrate the influence of that hyperparameter.
2. The authors partially address this concern: they argue that the use of languages as groups is natural in a multilingual setting and that other groups are outside the scope of the paper. They add to the revision a comment that CTC-DRO could be combined in future work with separate methods for discovering latent groups.
3. The authors partially address this concern in the revision and rebuttal. They emphasize that focusing on a smaller number of languages allows for a more rigorous evaluation and follows the experimental setup of the original group DRO paper, point out that Appendix H includes an 18-language experiment, and explain that scaling to 50-100 languages was not possible due to computational constraints.
4. This concern is partially addressed in the rebuttal and revision. The authors point out that they discuss in the paper the reasons that other fairness-promoting and loss-calibrated methods are not applicable to multilingual CTC training, and they enhance the discussion somewhat in the revision.
5. This concern is partially addressed in the rebuttal and revision. The authors argue that the methods mentioned by the reviewer are out of scope for the paper in that they examine "architectural and representation-level improvements to include language information" while the proposed CTC-DRO algorithm directly aims to improve worst-group performance.
6. This concern is addressed in the rebuttal and revision. The authors argue why length-normalizing the CTC loss is not appropriate and point out that Appendix G shows that this form of normalization leads to poor performance. That said, I think there is still an outstanding issue around length normalization. Namely, I think the authors should have considered a version of CTC-DRO in which the update to $\theta$ is based on the standard CTC loss while the update to $q_{g}$ is based on length-normalized CTC losses. This would probably have very similar effects to the length-matched batch sampler.
7. This concern is addressed in the rebuttal: the authors point out that the [-SMOOTH] condition still uses the group DRO weight update.
8. This concern is addressed in the rebuttal: the authors perform the same ablation study on langugage set 1 and show that both components of the CTC-DRO algorithm are needed for the best performance.
9. This concern is partially addressed in the rebuttal: the authors respond that CER is the primary metric of interest in the study, and that they report LID simply to remain consistent with standard practice for the ML-SUPERB task.
10. This concern is completely addressed in the rebuttal and revision: the authors report results showing that CTC-DRO does not significantly degrade performance on the best languages and they add this analysis to the revised manuscript.
11. This concern is addressed in the rebuttal and revision: the authors explain that Equation 10 was constructed to achieve the desired dynamics while being asymptotically equivalent to the original group DRO update as $\alpha$ approaches infinity.
12. This concern is addressed in the rebuttal and revision: the authors agreed with the reviewer's concern and updated the manuscript accordingly.

**Reviewer Scores:**

**UcsN** - I believe this reviewer would have increased their score, as their most substantive concern (analysis of hyperparameter sensitivity) was well addressed.

**sUDj** - I do not believe that this reviewer would have increased their score, but I also am downweighting the review because I do not think the reviewer fully understood the contribution of the paper.

**LdZ2** - This reviewer might have increased their score. The authors provided good responses to the reviewers concerns while also pushing back on some of them.

**id6Z** - I believe this reviewer would have increased their score, as many of their concerns were well addressed in the revision and rebuttal. I also consider this review to be the most thorough one that the paper received.

---

### Decision · Program_Chairs · 2026-01-26

Accept (Poster)